# Elevated glycolytic metabolism of monocytes limits the generation of HIF1A-driven migratory dendritic cells in tuberculosis

Mariano Maio[1,2,3], Joaquina Barros[1,2,3], Marine Joly[2,4], Zoi Vahlas[2,4], José Luis Marín Franco[1,2], Melanie Genoula[1,2], Sarah C Monard[2,4], María Belén Vecchione[3], Federico Fuentes[1], Virginia Gonzalez Polo[3], María Florencia Quiroga[3], Mónica Vermeulen[1], Thien-Phong Vu Manh[5], Rafael J Argüello[5], Sandra Inwentarz[6], Rosa Musella[6], Lorena Ciallella[6], Pablo González Montaner[6], Domingo Palmero[6], Geanncarlo Lugo Villarino[2,4], María del Carmen Sasiain[1,2], Olivier Neyrolles[2,4], Christel Vérollet[2,4†], Luciana Balboa[1,2,3*†]

[1]Instituto de Medicina Experimental (IMEX)-CONICET, Academia Nacional de Medicina, Buenos Aires, Argentina; [2]International Associated Laboratory (LIA) CNRS IM-TB/HIV (1167), Buenos Aires, Argentina / International Research Project Toulouse, Toulouse, France; [3]Instituto de Investigaciones Biomédicas en Retrovirus y Sida (INBIRS), Consejo Nacional de Investigaciones Científicas y Técnicas (CONICET) - Universidad de Buenos Aires, Buenos Aires, Argentina; [4]Institut de Pharmacologie et de Biologie Structurale, Université de Toulouse, CNRS, UPS, Toulouse, France; [5]Aix Marseille University, CNRS, INSERM, CIML, Centre d'Immunologie de Marseille-Luminy, Marseille, France; [6]Instituto Prof. Dr. Raúl Vaccarezza and Hospital de Infecciosas Dr. F.J. Muñiz, Buenos Aires, Argentina

*For correspondence:
luciana_balboa@hotmail.com

†These authors contributed equally to this work

Competing interest: The authors declare that no competing interests exist.

**Abstract** During tuberculosis (TB), migration of dendritic cells (DCs) from the site of infection to the draining lymph nodes is known to be impaired, hindering the rapid development of protective T-cell-mediated immunity. However, the mechanisms involved in the delayed migration of DCs during TB are still poorly defined. Here, we found that infection of DCs with *Mycobacterium tuberculosis* (Mtb) triggers HIF1A-mediated aerobic glycolysis in a TLR2-dependent manner, and that this metabolic profile is essential for DC migration. In particular, the lactate dehydrogenase inhibitor oxamate and the HIF1A inhibitor PX-478 abrogated Mtb-induced DC migration in vitro to the lymphoid tissue-specific chemokine CCL21, and in vivo to lymph nodes in mice. Strikingly, we found that although monocytes from TB patients are inherently biased toward glycolysis metabolism, they differentiate into poorly glycolytic and poorly migratory DCs compared with healthy subjects. Taken together, these data suggest that because of their preexisting glycolytic state, circulating monocytes from TB patients are refractory to differentiation into migratory DCs, which may explain the delayed migration of these cells during the disease and opens avenues for host-directed therapies for TB.

## eLife assessment

This **useful** study tests the hypothesis that monocytes purified from tuberculosis patients differentiate into dendritic cells with different migratory capacities. The authors conclude that these monocytes are metabolically pre-conditioned to differentiate, with reduced expression of Hif1a and a

glycolytically exhaustive phenotype, resulting in low migratory and immunologic potential. Overall, the evidence provided is **convincing**, advancing the field substantively and providing novel insights.

## Introduction

Tuberculosis (TB) remains a major global health problem, responsible for approximately 1.6 million deaths annually. The causative agent of TB, *Mycobacterium tuberculosis* (Mtb), is a highly successful pathogen that has evolved several strategies to weaken the host immune response. Although reliable immune correlates of protective immunity against Mtb are still not well-defined, it is widely accepted that Th1 cells contribute to protection by secreting IFN-γ and promoting antimycobacterial activity in macrophages (*Weiss and Schaible, 2015*). Importantly, the induction of a strong Th1 immune response relies on the generation of immunogenic dendritic cells (DCs) with strong migratory properties (*Khader et al., 2006*; *Wolf et al., 2007*; *Cooper, 2009*; *Lai et al., 2018*). Mtb has been shown to interfere with several DC functions, thus impairing the induction and development of adaptive immunity (*Urdahl et al., 2011*; *Chandra et al., 2022*; *Ernst, 2018*). For instance, we and others previously reported that Mtb-exposed DCs have low capacity for mycobacterial antigen presentation and stimulation of Mtb-specific CD4+ T cells (*Balboa et al., 2016*; *Wolf et al., 2008*; *Balboa et al., 2010*; *Harding and Boom, 2010*; *Srivastava et al., 2016*). Additionally, Mtb-infected DCs were reported to have an impaired ability to migrate to lymph nodes in vitro (*Roberts and Robinson, 2014*; *Rajashree et al., 2008*) and in vivo in murine models (*Wolf et al., 2007*; *Lai et al., 2018*); however, the underlying molecular mechanisms of these phenotypes and their relevance to the migratory activity of monocyte-derived DCs in TB patients remain unknown.

Rapid, directed migration of DCs toward secondary lymphoid organs requires essential changes at the cellular and molecular levels (*Currivan et al., 2022*). Relatedly, the metabolic state of DCs is complex and varies according to cell origin, differentiation, and maturation states, as well as local microenvironment, among other factors (*Basit et al., 2018*; *Wculek et al., 2019*; *Du et al., 2018*; *Møller et al., 2022*). Studies have reported that upon pathogen sensing the transcription factor hypoxia-inducible factor-1α (HIF1A) increases glycolysis, which promotes immunogenic functions of DCs, such as IL-12 production, costimulatory marker expression (*Everts and Pearce, 2014*), and cell migration (*Guak et al., 2018*; *Liu et al., 2019*; *Everts et al., 2014*). By contrast, it was shown that HIF1A represses the proinflammatory output of LPS-stimulated DCs and can inhibit DC-induced T-cell responses in other settings (*Lawless et al., 2017*). To reconcile these disparate roles for HIF1A, it has been proposed that the impact of metabolic pathway activation on DC functions varies among DC subsets (*Wculek et al., 2019*). To this point, most prior studies have been conducted using murine conventional DCs and plasmacytoid DCs (*Du et al., 2018*). Recently, with the implementation of high-dimensional techniques, it was demonstrated that distinct metabolic wiring is associated with individual differentiation and maturation stages of DCs (*Adamik et al., 2022*), highlighting the importance of defining the metabolic profile of specific subsets of DCs under physiological or pathological conditions (*Møller et al., 2022*). Given the key role of DCs in the host response to TB, it is thus crucial to investigate DC metabolism in the context of Mtb infection (*Kumar et al., 2019*).

We previously demonstrated that the TB-associated microenvironment, as conferred by the acellular fraction of TB patient pleural effusions, inhibits HIF1A activity, leading to a reduction in glycolytic and microbicidal phenotypes in macrophages (*Marín Franco et al., 2020*). Moreover, activation of HIF1A enhances Mtb control at early times post-infection in mouse models (*Baay-Guzman et al., 2018*), and this effect was associated with a metabolic switch of alveolar macrophages toward an M1-like profile (*Marín Franco et al., 2020*). Given that HIF1A activation promotes protection at early stages of Mtb infection and given its role as a key regulator of DC migration and inflammation (*Liu et al., 2021*), we hypothesized that HIF1A could affect the functionality of DCs in regulating the initiation and orchestration of the adaptive immune response to Mtb, a process known to be delayed upon Mtb infection (*Lai et al., 2018*; *Urdahl et al., 2011*). Here, we show that HIF1A-mediated glycolysis promotes DC activation and migration in the context of TB. Importantly, we report active glycolysis in monocytes from TB patients, which leads to poor glycolytic induction and migratory capacities of monocyte-derived DCs.

## Results

### Mtb impacts metabolism in human monocyte-derived DCs

To determine the impact of Mtb on the metabolism of human monocyte-derived DCs (Mo-DCs), we assessed metabolic parameters associated with glycolysis and mitochondrial changes upon Mtb stimulation or infection. Cells undergoing aerobic glycolysis are characterized by increased consumption of glucose and the production and release of lactate. We measured lactate release and glucose consumption in Mo-DCs stimulated for 24 hr with equivalent doses of either irradiated (iMtb) or viable Mtb. DCs treated with either iMtb or viable Mtb released increased levels of lactate and consumed more glucose than untreated DCs (*Figure 1A and B*). Consistently, both iMtb treatment and Mtb infection resulted in an increase in expression of the key glycolysis-activating regulator HIF1A at both mRNA and protein levels (*Figure 1C and D*). Expression of the gene encoding the glycolytic enzyme lactate dehydrogenase A (*LDHA*), which catalyzes the conversion of lactate to pyruvate, was also increased in iMtb-treated or Mtb-infected DCs (*Figure 1E*). In agreement with their enhanced glycolysis profile, DCs stimulated with iMtb or infected with viable Mtb had increased expression of the glucose transporter SLC2A1 (GLUT1) (*Freemerman et al., 2014*; *Figure 1F*). Of note, *LDHA* and *GLUT1* are HIF1A target genes, and their upregulation correlated with the increase in HIF1A expression upon Mtb stimulation. To assess changes in the mitochondria, we measured mitochondrial mass and morphology. We found a higher mitochondrial mass as well as larger individual mitochondria in iMtb-stimulated DCs compared to untreated DCs (*Figure 1G and H*). In contrast to the findings obtained upon iMtb stimulation, Mtb-infected DCs displayed a reduction in their mitochondrial mass (*Figure 1G*). This result indicates that although both Mtb-infected and iMtb-exposed DCs show a clear increase in their glycolytic activity, divergent responses are observed in terms of mitochondrial mass. Therefore, our data indicate that Mtb impacts the metabolism of Mo-DCs, leading to mitochondrial changes and triggering glycolysis-associated parameters.

### Mtb exposure shifts DCs to a glycolytic profile over oxidative phosphorylation

To further characterize the metabolic profile of DCs upon iMtb stimulation or Mtb infection, we next evaluated the metabolism of DCs at single-cell level using the SCENITH technology (*Argüello et al., 2020*). This method is based on a decrease in ATP levels that is tightly coupled with a decrease in protein synthesis and displays similar kinetics (*Argüello et al., 2020*). By treating the cells with glucose or mitochondrial respiration inhibitors, and measuring their impact on protein synthesis by puromycin incorporation via flow cytometry, glucose and mitochondrial dependences can be quantified. Two additional derived parameters such as 'glycolytic capacity' and 'fatty acid and amino acid oxidation (FAO and AAO) capacity' were also calculated. SCENITH technology revealed a lower reliance on oxidative phosphorylation (OXPHOS) in parallel with an increase in the glycolytic capacity of iMtb-stimulated (*Figure 2A and B*)**,** Mtb-infected DCs and even bystander DCs (those cells that are not directly infected but stand nearby) (*Figure 2C and D*). Since bystander DCs are not in direct association with Mtb (Mtb-RFP-DCs), soluble mediators induced in response to infection may be sufficient to trigger glycolysis even in uninfected cells. No differences were observed for glucose dependence and FAO and AAO capacity (*Figure 2A–D*). Additionally, we found no changes between the FAO dependency in Mtb-stimulated DCs in comparison to control cells when the FAO inhibitor (etomoxir) was used (*Figure 2—figure supplement 1*). For the case of iMtb-stimulated DCs, we also assessed the intracellular rates of glycolytic and mitochondrial ATP production using Seahorse technology. Bioenergetic profiles revealed that iMtb increased the rate of protons extruded over time, or proton efflux rate (PER), as well as the basal oxygen consumption rate (OCR) in Mo-DCs (*Figure 2E*). The measurements of basal extracellular acidification rate (ECAR) and OCR were used to calculate ATP production rate from glycolysis (GlycoATP) and mitochondrial OXPHOS (MitoATP). The ATP production rates from both glycolysis and mitochondrial respiration were augmented upon iMtb stimulation (*Figure 2F*). Similar to SCENITH results, the relative contribution of GlycoATP to overall ATP production was increased, while MitoATP contribution was decreased in iMtb-treated cells compared to untreated cells (*Figure 2G*). These results confirmed the change in DC metabolism induced by Mtb, with an increase in the relative glycolytic contribution to overall metabolism at the expense of the OXPHOS pathway. Together, metabolic profiling indicates that a metabolic switch toward aerobic glycolysis occurs in Mo-DCs exposed to Mtb.

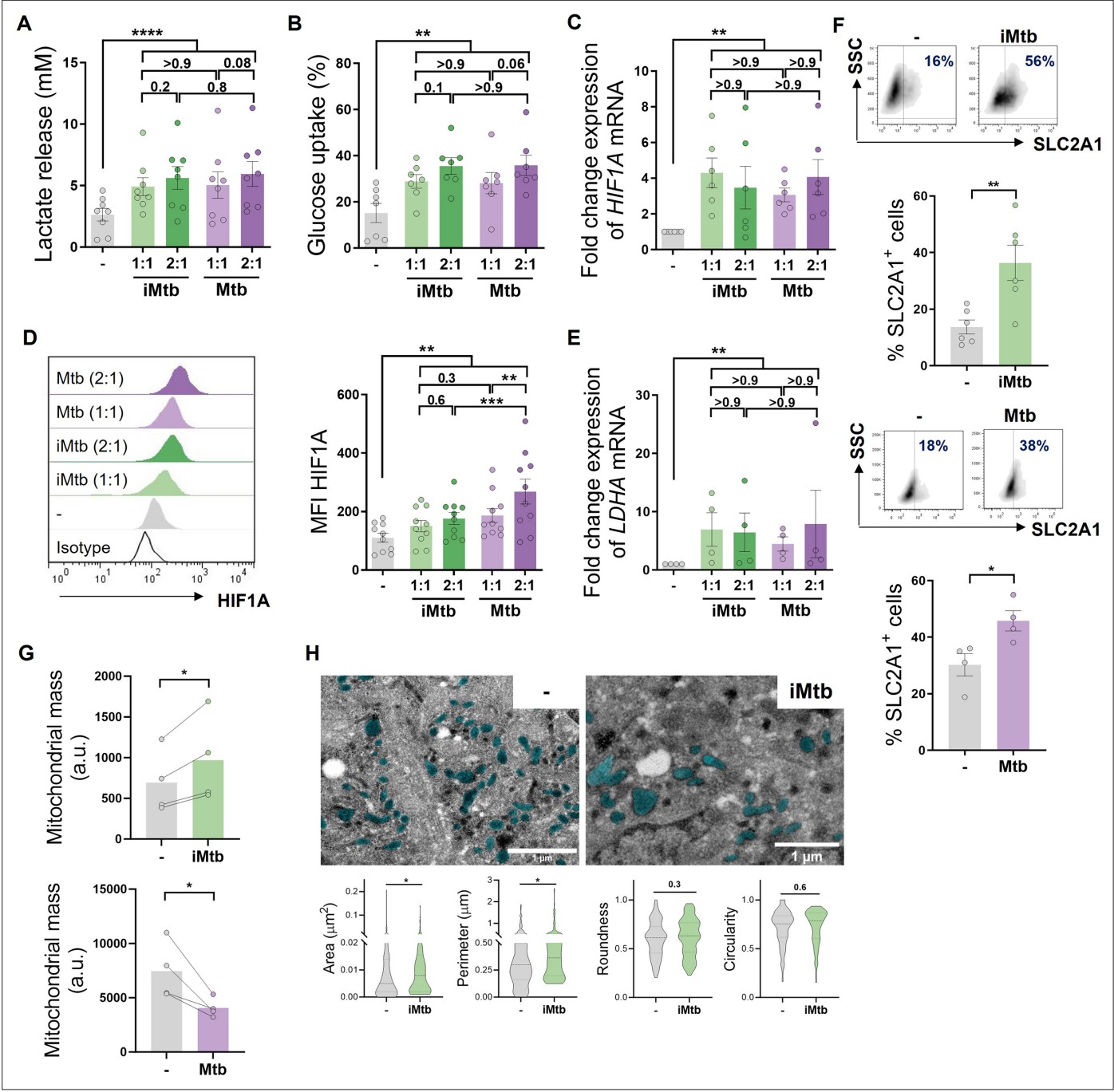

**Figure 1.** *Mycobacterium tuberculosis* (Mtb) rewires the metabolic network of monocyte-derived dendritic cells (Mo-DCs). Mo-DCs were stimulated with viable or irradiated Mtb (iMtb) at two multiplicities of infection (1 or 2 Mtb per DC) for 24 hr. Glycolysis was measured as (**A**) lactate release in culture supernatants (N = 8); (**B**) Glucose uptake measured in culture supernatants (N = 7); (**C**) relative expression of *HIF1A* mRNA normalized to *EEF1A1* control gene (N = 6). (**D**) Representative histograms of the mean fluorescence intensity (MFI) of HIF1A as measured by flow cytometry. Quantification shown in graph to the right (N = 10). (**E**) Relative expression of lactate dehydrogenase A (*LDHA*) mRNA normalized to *EEF1A1* control gene (N = 4). (**F**) FACS plots show the percentage of Glut1+ cells with and without iMtb stimulation or infected with viable Mtb in a representative experiment. Quantification of Glut1+ cells plotted below (N = 4–6). (**G**) MFI of Mitospy probe as a measurement of mitochondrial mass for Mo-DCs treated (or not) with iMtb (upper panel) or infected with viable Mtb (lower panel). The data are represented as scatter plots, with each circle representing a single individual, means ± SEM are shown (N = 4). (**H**) Representative electron microscopy micrographs of control and iMtb-stimulated DCs showing mitochondria colored in cyan (left panels) and quantified morphometric analysis (right panels) (N = 4). Statistical significance was assessed in (**A–E**) using two-way ANOVA followed by Tukey's multiple-comparisons test (*p<0.05; **p<0.01; ****p<0.0001), and in (**F–H**) using paired *t*-test (*p<0.05) for iMtb versus controls. All values are expressed as means ± SEM.

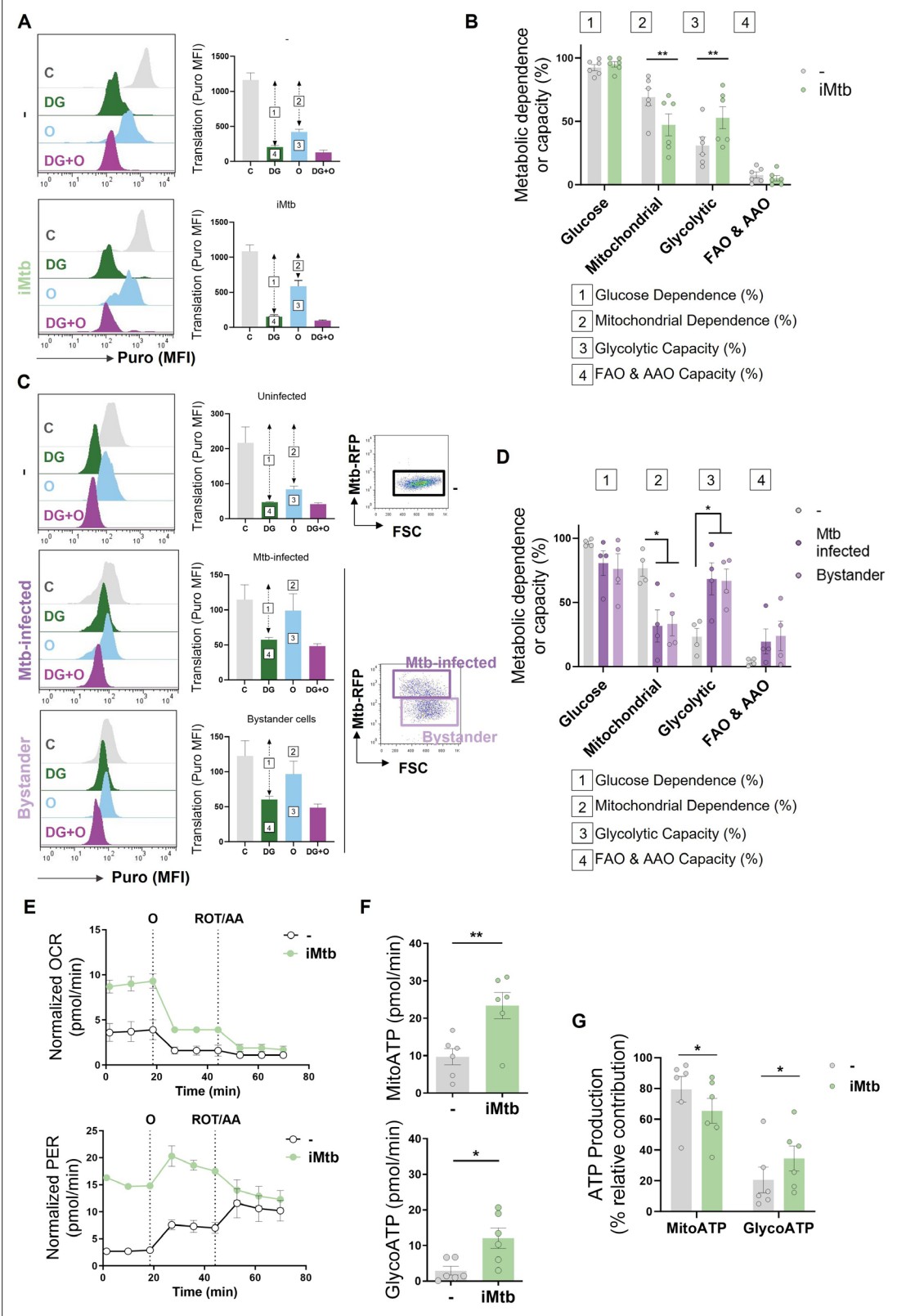

**Figure 2.** *Mycobacterium tuberculosis* (Mtb) skews dendritic cell (DC) metabolism toward glycolysis. Monocyte-derived DCs (Mo-DCs) were stimulated with irradiated Mtb (iMtb) or infected with Mtb expressing red fluorescent protein (Mtb-RFP, panel **C**). (**A**) Representative histograms showing the translation level after puromycin (Puro) incorporation and staining with a monoclonal anti-Puro (anti-Puro mean fluorescence intensity [MFI]) in response to inhibitor treatment (C, control; DG, 2-deoxy-D-glucose; oligomycin, O; or combination treatment, DG + O). The bar plots show the values of the anti-

*Figure 2 continued on next page*

*Figure 2 continued*

Puro MFI from six donors. Arrows and numbers inside boxes denote the differences between the MFI of Puro in the different treatments that are used to calculate the glucose dependence (1) and fatty acids and amino acids oxidation (FAO and AAO) capacity (4); and the mitochondrial dependency (2) and glycolytic capacity (3). (**B**) Relative contributions of glycolytic and FAO and AAO capacities and glucose and mitochondrial dependences to overall DC metabolism analyzed with SCENITH (N = 6). (**C, D**) DCs were infected with Mtb-RFP for 24 hr, thereafter the metabolic profile was evaluated using SCENITH. (**C**) Representative histograms showing the translation level after Puro incorporation are shown for uninfected, Mtb-infected and bystander DCs (those cells that are not infected directly but rather stand nearby). The bar plots show the values of the anti-Puro MFI from four donors. Right panel shows representative plots showing the gating strategy to distinguish the populations within Mtb-infected cultures, which includes RFP⁺ (Mtb-infected DCs) and RFP⁻ (bystander DCs) cells. (**D**) Relative contributions of glycolytic and FAO and AAO capacities and glucose and mitochondrial dependences to DC metabolism (N = 4). (**E**) Kinetic profile of proton efflux rate (PER; lower panel) and oxygen consumption rate (OCR; upper panel) measurements in control and iMtb-stimulated DCs in response to inhibitor treatments (oligomycin, O; ROT/AA, rotenone/antimycin A), obtained using an Agilent Seahorse XFe24 Analyzer. PER and OCR measurements were normalized to the area covered by cells. (**F**) ATP production rate from mitochondrial oxidative phosphorylation (MitoATP) and glycolysis (glycoATP). MitoATP production rate and glycoATP production rate were calculated from OCR and ECAR measurements in control and iMtb-stimulated DCs (N = 6). (**G**) Percentages of MitoATP and GlycoATP relative to overall ATP production (N = 6). Statistics in (**B, F–G**) are from paired *t*-test (*p<0.05; **p<0.01) for iMtb versus controls. Statistics in (**D**) are two-way ANOVA followed by Tukey's multiple-comparisons test (*p<0.05) as depicted by lines. The data are represented as scatter plots, with each circle representing a single individual, means ± SEM are shown.

The online version of this article includes the following figure supplement(s) for figure 2:

**Figure supplement 1.** Contribution of the fatty acid oxidation (FAO) to dendritic cell (DC) metabolism in response to irradiated *Mycobacterium tuberculosis* (iMtb).

## Mtb triggers the glycolytic pathway through TLR2 ligation

Since Mtb is sensed by Toll-like receptors (TLR)-2 and -4 (*Quesniaux et al., 2004*), we investigated the contribution of these receptors to glycolysis activation in Mo-DCs upon Mtb stimulation. Using specific neutralizing antibodies for these receptors, we found that TLR2 ligation, but not that of TLR4, was required to trigger the glycolytic pathway, as reflected by a decrease in lactate release, glucose consumption, and HIF1A expression in iMtb-stimulated DCs treated with an anti-TLR2 antibody (*Figure 3A–C*). As a control, and as expected given the reliance on TLR4 for LPS sensing (*Chow et al., 1999*), lactate release and glucose consumption were abolished in LPS-stimulated DCs in the presence of neutralizing antibodies against TLR4 but not TLR2 (*Figure 3—figure supplement 1A and B*). Moreover, blockade of TLR2 also diminished glycolytic ATP production in iMtb-stimulated DCs without altering OXPHOS-associated ATP production (*Figure 3D*) or the size and morphology of mitochondria (*Figure 3—figure supplement 1C*), suggesting that TLR2 engagement by iMtb is required for the induction of glycolysis but not mitochondrial respiration. Interestingly, TLR2 ligation was also necessary for lactate release and HIF1A upregulation triggered by viable Mtb, (*Figure 3—figure supplement 1C and D*). To further confirm the involvement of TLR2 in the induction of glycolysis, we tested the effect of synthetic (Pam₃CSK₄) or mycobacterial (peptidoglycans, PTG) TLR2 agonists (*Underhill et al., 1999*; *Schwandner et al., 1999*) and found that both ligands induced lactate release and glucose consumption in DCs (*Figure 3E and F*), without affecting cell viability (*Figure 3—figure supplement 1F*). Thus, our data indicate that Mtb induces glycolysis in Mo-DCs through TLR2 engagement.

## HIF1A is required for DC maturation upon iMtb stimulation but not for CD4⁺ T lymphocyte polarization

To determine the impact of glycolysis on DC maturation and the capacity to activate T cells, we inhibited HIF1A activity in iMtb-stimulated DCs employing two HIF1A inhibitors that display different mechanisms of action. The first is PX-478 (PX) that lowers HIF1A levels by inhibiting HIF-1α deubiquitination, decreases *HIF1A* mRNA expression, and reduces *HIF1A* translation *Koh et al., 2008*; the second one is echinomycin (Ech), which inhibits the binding of HIF1A to the hypoxia response element thereby blocking HIF1A DNA binding capability (*Cairns et al., 2007*; *Kong et al., 2005*). Treatment with either HIF1A inhibitor PX or Ech diminished lactate release and glucose consumption in iMtb-stimulated DCs without affecting cell viability at the indicated concentration (*Figure 4—figure supplement 1A–F*). HIF1A inhibition by PX significantly abolished ATP production associated with glycolysis without affecting absolute levels of OXPHOS-derived ATP production in iMtb-stimulated DCs (*Figure 4A*, *Figure 4—figure supplement 1G*). In line with these results, the glycolytic capacity was reduced in

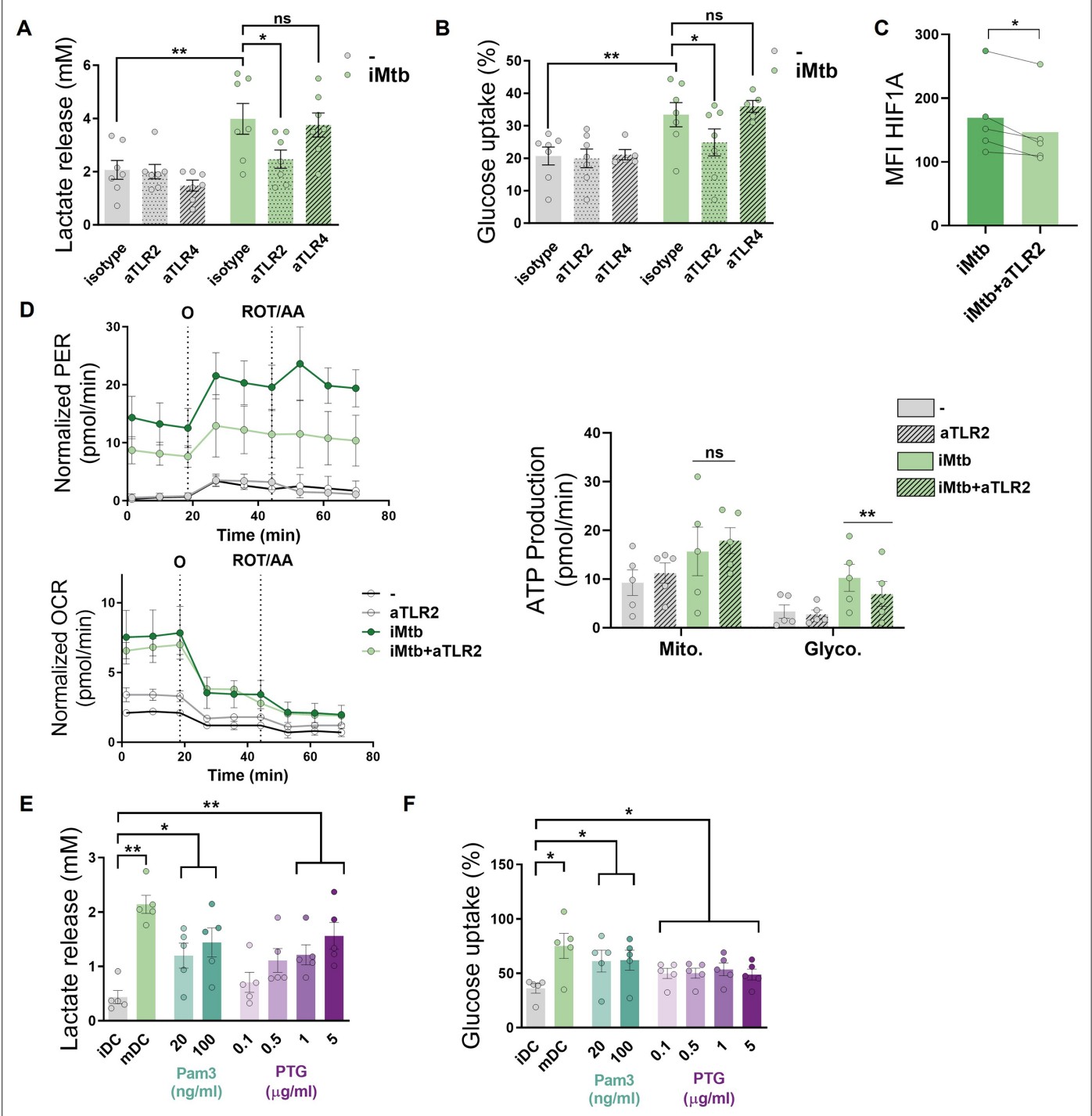

**Figure 3.** *Mycobacterium tuberculosis* (Mtb) triggers glycolysis through TLR2 ligation in monocyte-derived dendritic cells (Mo-DCs). Mo-DCs were stimulated with irradiated Mtb (iMtb) in the presence of neutralizing antibodies against either TLR2 (aTLR2), TLR4 (aTLR4), or their respective isotype controls. (**A**) Lactate release as measured in supernatant (N = 7). (**B**) Glucose uptake as measured in supernatant (N = 7). (**C**) Mean fluorescence intensity (MFI) of HIF1A as measured by flow cytometry (N = 4). (**D**) Kinetic profile of proton efflux rate (PER) and oxygen consumption rate (OCR) measurements (left panels). Metabolic flux analysis showing quantification of mitochondrial ATP production and glycolytic ATP production (right panel) (N = 5). (**E, F**) Mo-DCs were stimulated with Pam3Cys or Mtb peptidoglycan (PTG) at the indicated concentrations. (**E**) Lactate release as measured in supernatant (N = 5). (**F**) Glucose uptake as measured in supernatant (N = 5). Statistics in (**A–B, E–F**) are two-way ANOVA followed by Tukey's multiple-comparisons test (*p<0.05; **p<0.01; ****p<0.0001). Statistics in (**C, D**) are from paired *t*-test (*p<0.05) for iMtb versus controls. The data are represented as scatter plots, with each circle representing a single individual, means ± SEM are shown.

*Figure 3 continued on next page*

*Figure 3 continued*

The online version of this article includes the following figure supplement(s) for figure 3:

**Figure supplement 1.** TLR2 ligation triggers glycolysis in monocyte-derived dendritic cells (Mo-DCs).

iMtb-stimulated DCs treated with Ech (*Figure 4—figure supplement 1H*). Although HIF1A inhibitors did not affect the uptake of iMtb by DCs (*Figure 4—figure supplement 1I*), we observed a reduction in the expression of activation markers CD83 and CD86, but not in the inhibitory molecule PD-L1, upon treatment with PX (*Figure 4B*) or Ech (*Figure 4—figure supplement 2*). We then measured cytokine production in iMtb-stimulated DCs after HIF1A inhibition and noted a reduction in TNF-α and an increase in IL-10 production by PX-treated cells (*Figure 4C*). To assess the capacity of DCs to activate T cells in response to mycobacterial antigens, we co-cultured DCs and autologous CD4+ T cells from PPD+ donors in the presence or absence of HIF1A inhibitors and measured the overall IFN-γ and IL-17 production in culture supernatants as well as cell surface expression of T cell markers. We found no significant differences in the activation profile of autologous CD4+ T cells in coculture with iMtb-stimulated DCs treated or not with PX (*Figure 4D and E*, *Figure 4—figure supplement 3*). We conclude that while HIF1A is important for the maturation of iMtb-stimulated Mo-DCs, it does not influence their capacity to activate CD4+ T cells, at least in vitro.

## HIF1A-mediated glycolysis triggers the motility of DCs upon iMtb stimulation

Since DC migration to lymph nodes is essential to initiate an adaptive immune response and glycolytic activity has been reported to control DC migration upon stimulation (*Guak et al., 2018*; *Liu et al., 2019*), we evaluated the migratory properties of iMtb-stimulated DCs in the presence of inhibitors of HIF1A and LDH which catalyzes the interconversion of pyruvate and lactate. First, we confirmed that PX and oxamate (OX), a well-established LDH inhibitor, diminished the glycolytic activity of iMtb-stimulated human Mo-DCs, as demonstrated by reduced lactate release (*Figure 5A*, *Figure 4—figure supplement 1A*). Next, using a transwell migration assay, we found that PX and OX treatment significantly diminished the chemotactic activity of iMtb-stimulated human Mo-DCs in response to CCL21 (*Figure 5B*), a CCR7 ligand responsible for the migration of DCs into lymphoid organs. We also assessed the three-dimensional (3D) migration capacity of iMtb-stimulated DCs through a collagen matrix in which DCs use an amoeboid migration mode (*Cougoule et al., 2018*) and found that 3D migration was significantly impaired upon HIF1A or glycolysis inhibition (*Figure 5C*). The role of glycolysis in the migration of iMtb-stimulated Mo-DCs was further confirmed using an additional LDHA inhibitor, GSK2837808A, which reduced both the release of lactate by iMtb-stimulated Mo-DCs and their migration in response to CCL21 (*Figure 5—figure supplement 1A and B*). Attenuation of cell migration through collagen induced by OX and PX was also confirmed in Mtb-infected DCs (*Figure 5D*). To further investigate the effects of glycolysis on cell migration, we turned to an in vivo model. Murine bone marrow-derived DCs (BMDCs) isolated and stimulated with iMtb in the presence or absence of PX or OX were labeled with CFSE and transferred into naïve mice (*Figure 5E*). Similar to human Mo-DCs, iMtb stimulation increased glycolysis in BMDCs, which was inhibited by PX and OX treatment in vitro (*Figure 5—figure supplement 1C*). Three hours after the transfer of BMDCs into recipient mice, nearby lymph nodes were collected for DC quantification (*Figure 5E*). A higher number of adoptively transferred DCs (CFSE-labeled CD11c+ cells) were detected in lymph nodes from mice that received iMtb-stimulated BMDCs compared to mice that received untreated BMDCs or iMtb-BMDCs treated with either PX or OX (*Figure 5F*, *Figure 5—figure supplement 1D*). Of note, we verified that CCR7 expression on iMtb-stimulated BMDCs was not affected by OX or PX treatment, so the effect could not be ascribed to downregulation of the chemokine receptor (*Figure 5—figure supplement 1E*). Therefore, we conclude that HIF1A-mediated glycolysis is required for the successful migration of iMtb-stimulated DCs into lymph nodes.

## Stabilization of HIF1A promotes migration of tolerogenic DCs and DCs derived from TB patient monocytes

Since DC differentiation is skewed, at least partially, toward a tolerogenic phenotype during TB (*Balboa et al., 2013*; *Parlato et al., 2018*; *Sakhno et al., 2015*), we investigated whether tolerogenic

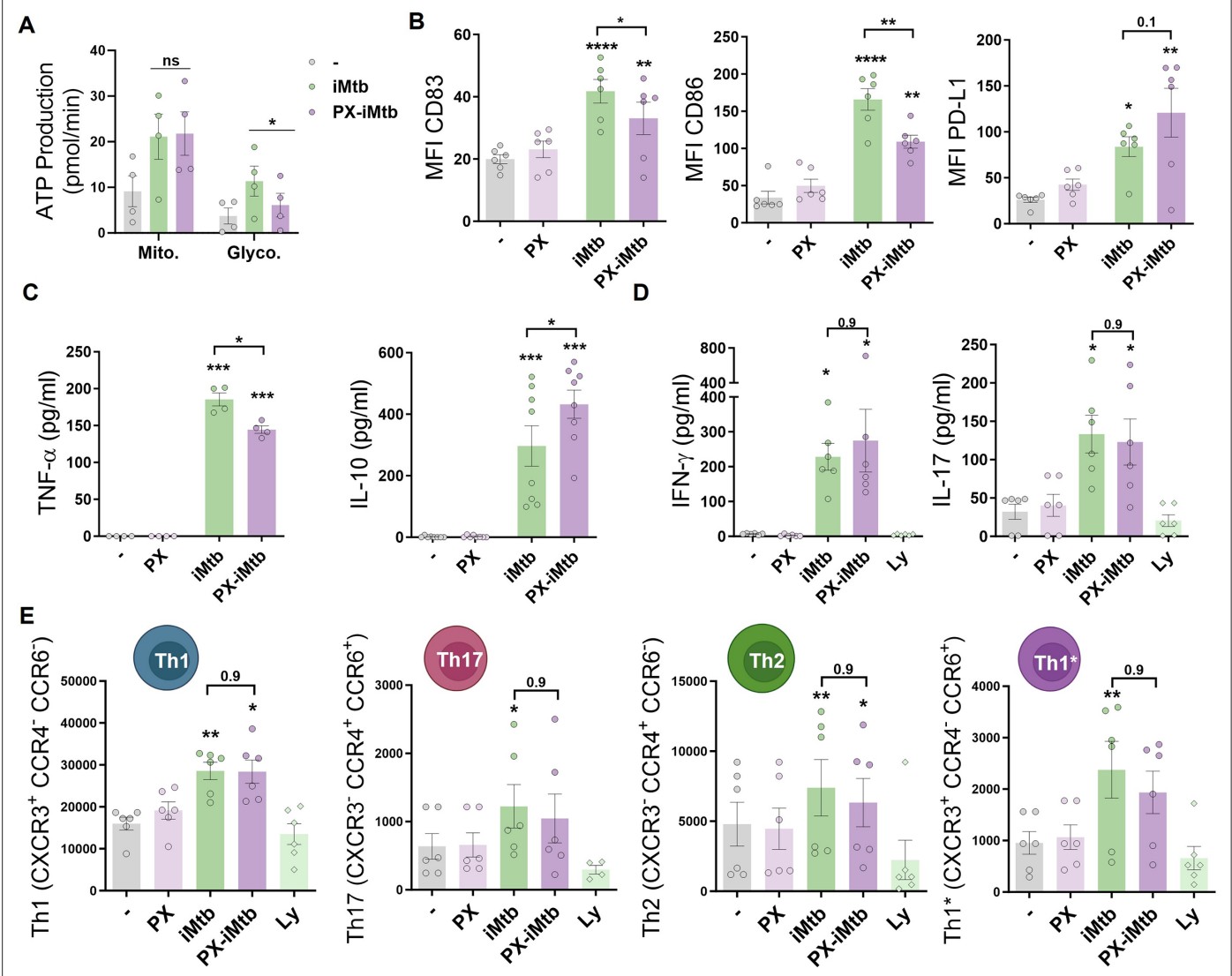

**Figure 4.** HIF1A is required for dendritic cell (DC) maturation upon irradiated *Mycobacterium tuberculosis* (iMtb) stimulation but not for CD4+ T lymphocyte polarization. (**A–C**) Monocyte-derived DCs (Mo-DCs) were stimulated with iMtb in the presence or absence of the HIF1A inhibitor PX-478 (PX). (**A**) Metabolic flux analysis showing quantification of mitochondrial ATP production and glycolytic ATP production, as in *Figure 2G* (N = 4). (**B**) Mean fluorescence intensity (MFI) of CD83, CD86, and PD-L1 as measured by flow cytometry (N = 6). (**C**) TNF-α and IL-10 production by Mo-DCs measured by ELISA (N = 4–8). (**D, E**) Monocytes from PPD+ healthy donors were differentiated toward DCs, challenged or not with iMtb in the presence or absence of PX for 24 hr, washed, and co-cultured with autologous CD4+ T cells for 5 days. (**D**) Extracellular secretion of IFN-γ and IL-17 as measured by ELISA (N = 6). (**E**) Absolute abundance of Th1, Th17, Th2, and Th1/Th17 CD4+ T cells after coculture with DCs (N = 6). When indicated, lymphocytes without DCs were cultured (Ly). Statistical significance based on two-way ANOVA followed by Tukey's multiple-comparison test (*p<0.05; **p<0.01). The data are represented as scatter plots, with each circle representing a single individual, means ± SEM are shown.

The online version of this article includes the following figure supplement(s) for figure 4:

**Figure supplement 1.** HIF1A activity is required to trigger the glycolytic pathway in irradiated *Mycobacterium tuberculosis* (iMtb)-stimulated dendritic cells (DCs).

**Figure supplement 2.** HIF1A is required to adopt a mature phenotype in irradiated *Mycobacterium tuberculosis* (iMtb)-stimulated dendritic cells (DCs).

**Figure supplement 3.** Gating strategy to define CD4+ T cells in response to *Mycobacterium tuberculosis* (Mtb).

DCs can be reprogrammed into immunogenic DCs by modulating their glycolytic pathway after iMtb stimulation. To this end, we generated tolerogenic Mo-DCs by adding dexamethasone (Dx) before stimulation with iMtb in the presence or absence of dimethyloxalylglycine (DMOG), which stabilizes the expression of HIF1A. HIF1A expression is tightly regulated by prolyl hydroxylase domain containing proteins, which facilitate the recruitment of the von Hippel-Lindau (VHL) protein, leading

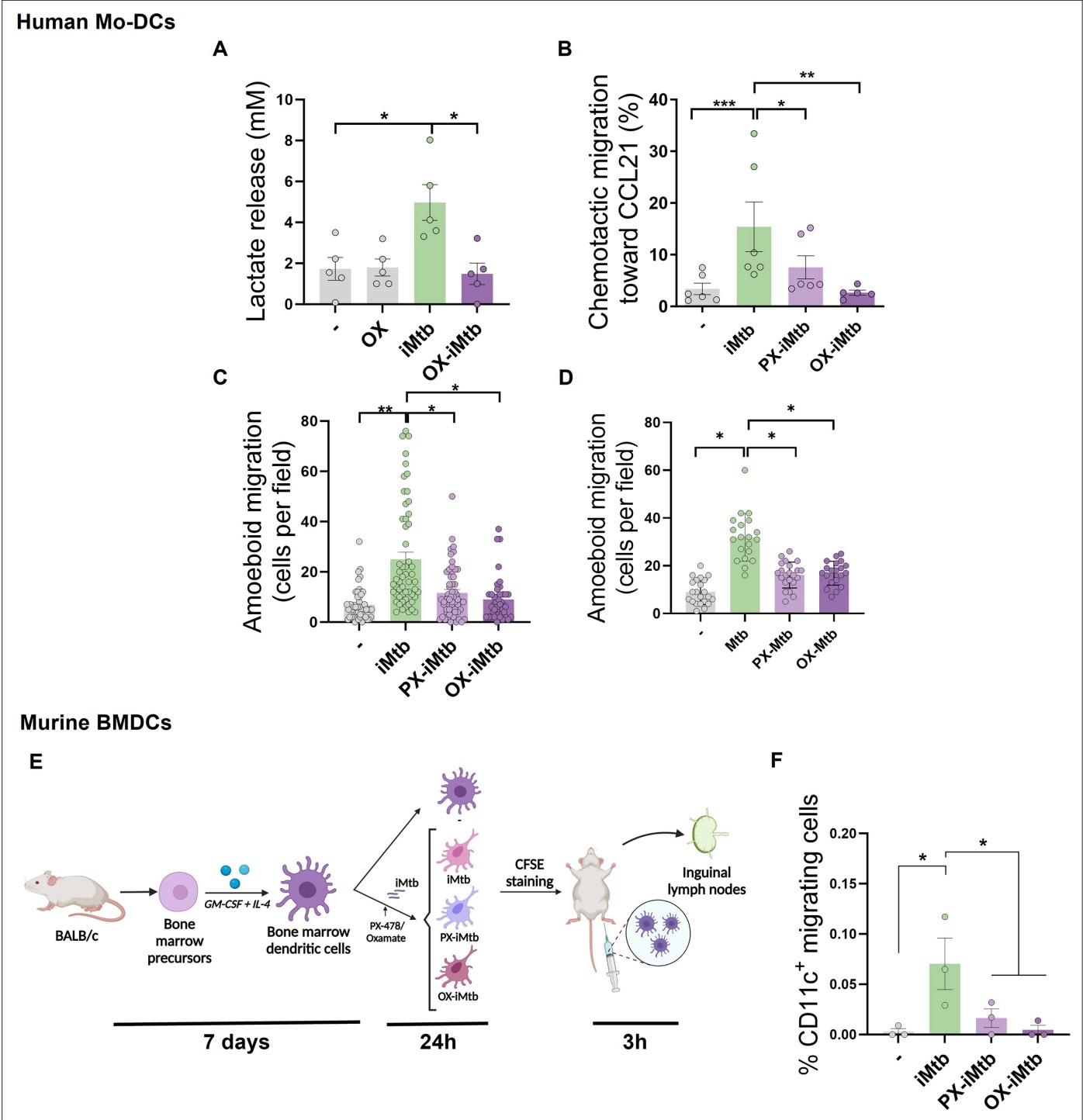

**Figure 5.** HIF1A-mediated-glycolysis is required to trigger migratory activity in irradiated *Mycobacterium tuberculosis* (iMtb)-stimulated dendritic cells (DCs). Monocyte-derived DCs (Mo-DCs) were treated (or not) with HIF1A inhibitor PX-478 (PX) or LDH inhibitor oxamate (OX) and stimulated with iMtb for 24 hr. (**A**) Lactate release as measured in supernatants in DCs stimulated or not with iMtb in the presence of OX (N = 5). (**B**) Percentage of migrated cells toward CCL21 relative to the number of initial cells per condition (N = 6). (**C, D**) Three-dimensional amoeboid migration of DCs through a collagen matrix after 24 hr. Cells within the matrix were fixed and stained with DAPI. Images of the membrane of each insert were taken and the percentage of cells per field were counted. (**C**) Mo-DCs stimulated with iMtb for 24 hr (N = 5). (**D**) Mo-DCs infected with Mtb for 24 hr (N = 4). The data are represented as scatter plots, where each circle represents a microphotograph sourced from either five (**C**) or four (**D**) independent donors, with each experiment typically including between 5 to 10 microphotographs. (**E**) Representative schematic of the experimental setup for in vivo migration assays. (**F**) Percentages of migrating bone marrow-derived DCs (BMDCs) (CFSE-labeled among CD11c+) recovered from inguinal lymph nodes (N = 3).

*Figure 5 continued on next page*

*Figure 5 continued*

Statistical significance assessed by (**A, B**) ANOVA followed by Dunnett's multiple-comparisons test (*p<0.05; **p<0.01); (**C, D**) Nested ANOVA followed by Dunnett's multiple-comparisons test (*p<0.05; **p<0.01); (**F**) ANOVA followed by Holm–Sidak's multiple-comparisons test (*p<0.05).

The online version of this article includes the following figure supplement(s) for figure 5:

**Figure supplement 1.** Glycolysis is required to trigger the migratory activity in irradiated *Mycobacterium tuberculosis* (iMtb)-stimulated dendritic cells (DCs).

to ubiquitination and degradation of HIF1A by the proteasomes (*McGettrick and O'Neill, 2020*). DMOG inhibits the prolyl hydroxylase domain-containing proteins. Acquisition of the tolerogenic phenotype was confirmed by the lack of upregulation of costimulatory markers CD83 and CD86, as well as by increased PD-L1 expression in iMtb-DCs treated with Dx compared to control iMtb-DCs (*Figure 6—figure supplement 1A*). Moreover, Dx-treated DCs did not exhibit an increase in lactate release, consumption of glucose, or induction of HIF1A expression in response to iMtb, showing a high consumption of levels of glucose under basal conditions (*Figure 6A and B*). Of note, HIF1A stabilization using DMOG restored the HIF1A expression and lactate production in response to iMtb in Dx-treated DCs and increased the consumption of glucose (*Figure 6A and B*). Activation of HIF1A also improved 3D amoeboid migration, as well as 2D migration capacity of DCs toward CCL21 of iMtb-stimulated Dx-treated DCs (*Figure 6C and D*, *Figure 6—figure supplement 1B*). Confirming the relevance of these findings to human TB patients, we found that iMtb-stimulated Mo-DCs from TB patients were deficient in their capacity to migrate toward CCL21 (*Figure 6E*) and in glycolytic activity compared to Mo-DCs from healthy subjects (*Figure 6F and G*). Strikingly, stabilizing HIF1A expression using DMOG in Mo-DCs from TB patients restored their chemotactic activity in response to iMtb (*Figure 6H*). These data indicate that the impaired migratory capacity of iMtb-stimulated tolerogenic DCs or TB patient-derived DCs can be restored via HIF1A stabilization; thus, glycolysis is critical for DC function during TB in both murine and human contexts.

## CD16$^+$ monocytes from TB patients show increased glycolytic capacity

Since we observed differences in the metabolic activity of DCs derived from monocytes of TB patients when compared to healthy donors, we next focused on evaluating the release of lactate by DC precursors from both subject groups during the first hours of DC differentiation with IL-4/GM-CSF. We found a high release of lactate by monocytes from TB patients compared to healthy donors after 1 hr of differentiation (*Figure 7A*). Lactate accumulation increased in both subject groups after 24 hr with IL-4/GM-CSF (*Figure 7A*). Based on these differential glycolytic activities displayed by DC precursors from both subject groups at very early stages of the differentiation process, we decided to evaluate the ex vivo metabolic profile of monocytes using SCENITH. To this end, we assessed the baseline glycolytic capacity of the three main populations of monocytes: classical (CD14$^+$CD16$^-$), intermediate (CD14$^+$CD16$^+$), and non-classical (CD14$^{dim}$CD16$^+$) monocytes. We found that both populations of CD16$^+$ monocytes from TB patients had a higher glycolytic capacity than monocytes from healthy donors (*Figure 7B*). Moreover, the glycolytic capacity of CD16$^+$ monocytes (CD14$^+$CD16$^+$ and CD14$^{dim}$CD16$^+$) correlates with time since the onset of TB-related symptoms (*Figure 7C*), with no association to the extent or severity of lung disease (unilateral/bilateral lesions and with/without cavities, *Figure 7—figure supplement 1*). To further expand the metabolic characterization of monocyte subsets from TB patients, we used previously published transcriptomic data (GEO accession number: GSE185372) of CD14$^+$CD16$^-$, CD14$^+$CD16$^+$, and CD14$^{dim}$CD16$^+$ monocytes isolated from individuals with active TB, latent TB (IGRA$^+$), as well as from TB-negative healthy controls (IGRA$^-$) (*Hillman et al., 2022*). Within this framework, we performed high-throughput GeneSet Enrichment Analysis (GSEA) using the BubbleMap module of BubbleGUM, which includes a multiple testing correction step to allow comparisons between the three monocyte subsets (*Spinelli et al., 2015*). As expected, this approach reveals enrichments in genes associated with interferon responses (alpha and gamma) in patients with active TB compared to healthy donors (either IGRA$^-$ or latent TB) for all three monocyte subsets (*Figure 7D*). Consistent with our findings, glycolysis increases in active TB in both CD14$^+$CD16$^+$ and CD14$^{dim}$CD16$^+$ monocytes (albeit not significant), while it appears to decrease in classical CD14$^+$CD16$^-$ monocytes (*Figure 7D*). Unlike CD14$^+$CD16$^-$ cells, the inflammatory response is notably enriched in CD14$^+$CD16$^+$ and CD14$^{dim}$CD16$^+$ monocytes from patients with active TB compared those with latent TB or healthy subjects (*Figure 7D*), suggesting that their glycolytic profile correlates with a higher

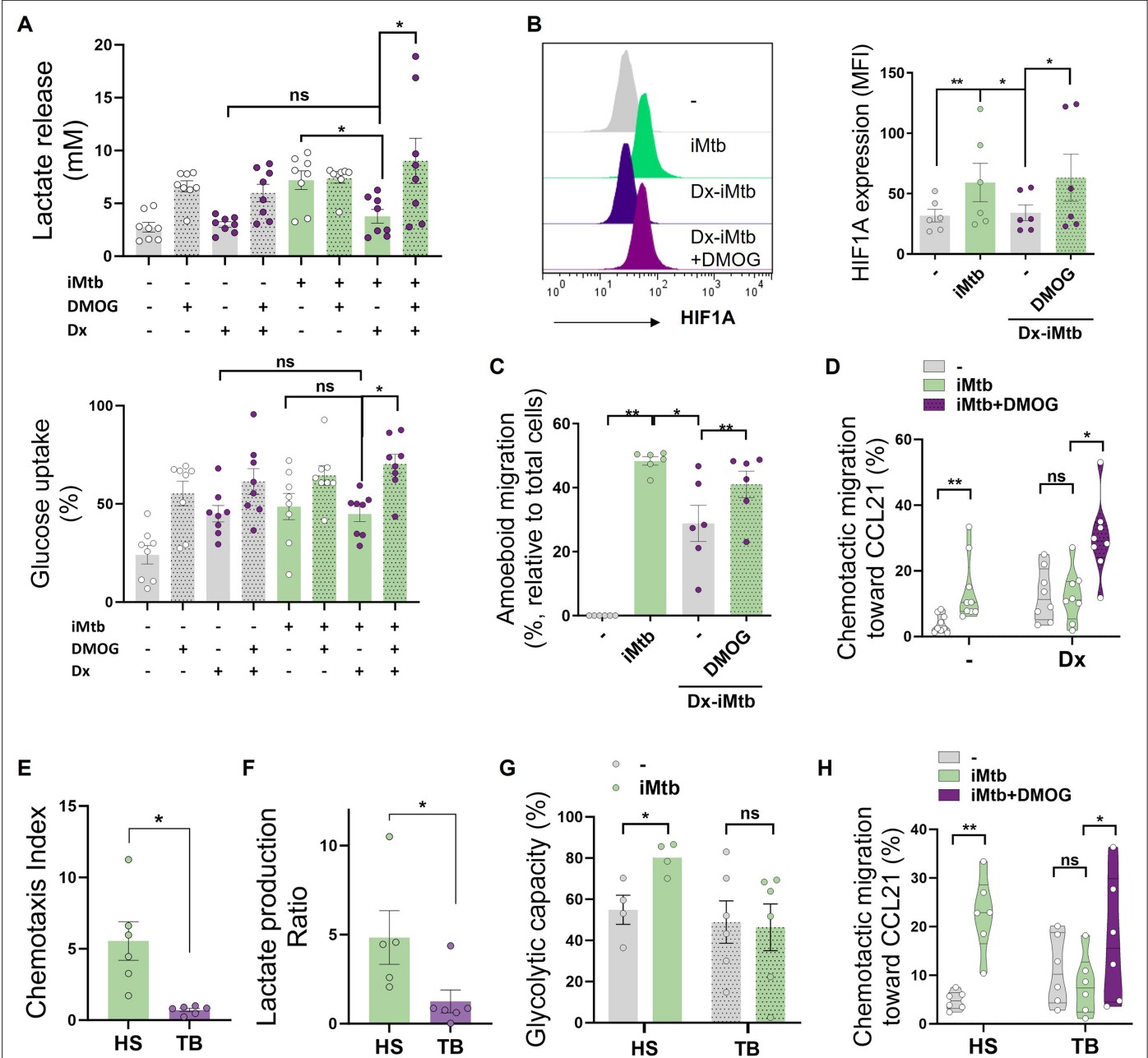

**Figure 6.** Stabilization of HIF1A promotes migration of tolerogenic dendritic cells (DCs) and monocyte-derived DCs (Mo-DCs) from tuberculosis (TB) patients. Tolerogenic Mo-DCs were generated by dexamethasone (Dx) treatment and were stimulated (or not) with irradiated *Mycobacterium tuberculosis* (iMtb) in the presence or absence of HIF1A activator dimethyloxalylglycine (DMOG). (**A**) Lactate release and glucose uptake as measured in supernatant (N = 8). (**B**) Mean fluorescence intensity (MFI) of HIF1A. Representative histograms and quantification are shown (N = 6). (**C**) Three-dimensional amoeboid migration of DCs through a collagen matrix. After 24 hr of migration, images of stacks within the matrix were taken every 30 μm. Percentage of migrating cells was defined as cells in the stacks within the matrix relative to total number of cells (N = 6). (**D**) Chemotactic activity toward CCL21 in vitro (N = 6). (**E–H**) Mo-DCs were generated from healthy subjects (HS) or TB patients, and DCs were stimulated (or not) with iMtb. (**E**) Chemotaxis index toward CCL21 (relative to unstimulated DCs) (N = 6). (**F**) Lactate production ratio relative to unstimulated DCs (N = 6). (**G**) Glycolytic capacity assessed by SCENITH (N = 4). (**H**) Chemotactic activity toward CCL21 of Mo-DCs from TB patients stimulated with iMtb and treated or not with DMOG (N = 6). Statistical significance assessed by (**A–D**) two-way ANOVA followed by Tukey's multiple-comparisons test (*p<0.05; **p<0.01); (**E–G**) unpaired *t*-test (*p<0.05); (**H**) paired *t*-test (*p<0.05). The data are represented as scatter plots, with each circle representing a single individual, means ± SEM are shown.

The online version of this article includes the following figure supplement(s) for figure 6:

**Figure supplement 1.** Profile of tolerogenic dendritic cells (DCs) induced by dexamethasone (Dx).

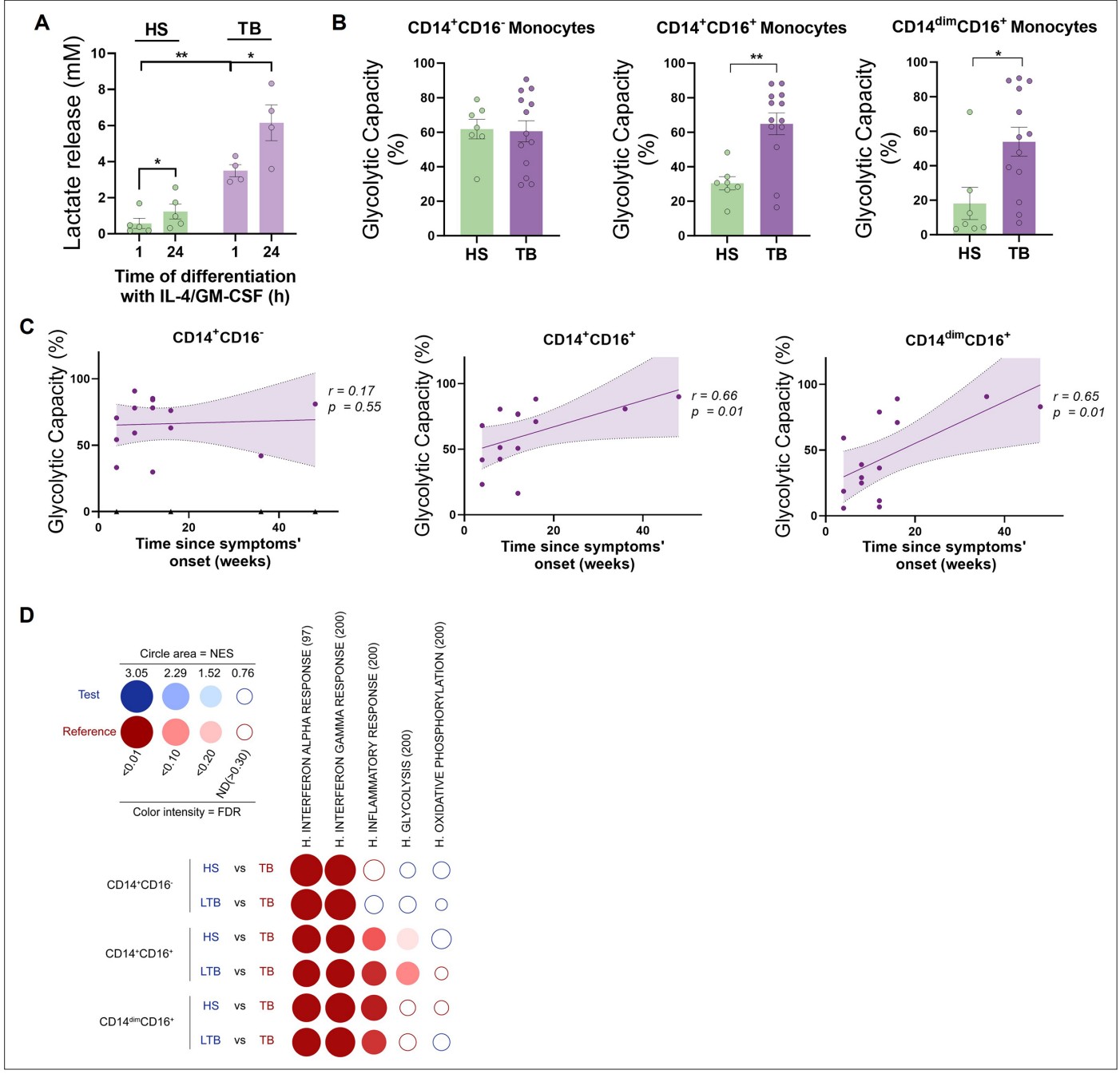

**Figure 7.** CD16+ monocytes from tuberculosis (TB) patients show increased glycolytic capacity. (**A**) Monocytes from TB patients or healthy subjects (HS) were isolated and cultured with IL-4 and GM-CSF for 24 hr. Accumulation of lactate in culture supernatants were measured at 1 and 24 hr of differentiation (N = 5). (**B**) Glycolytic capacity measured using SCENITH of monocyte subsets as defined by their CD14 and CD16 expression from HS and TB patients (N = 7). (**C**) Correlation analysis between the baseline glycolytic capacity and the evolution time of TB symptoms for each monocyte subset (CD14+CD16-, CD14+CD16+, and CD14dimCD16+, N = 14). Linear regression lines are shown. Spearman's rank test. The data are represented as scatter plots, with each circle representing a single individual, means ± SEM are shown. (**D**) BubbleMap analysis, a high-throughput extension of GeneSet Enrichment Analysis (GSEA), on the pairwise comparisons of monocytes from HS or donors with latent TB (LTB) vs. patients with active TB (TB), for each monocyte subset (CD14+CD16-, CD14+CD16+, and CD14dimCD16+). The gene sets shown come from the Hallmark (**H**) collection of the Molecular Signature Database (MSigDB). The colors of the BubbleMap correspond to the population from the pairwise comparison in which the geneset is enriched (red if geneset is enriched in TB). The bubble area is proportional to the GSEA normalized enrichment score (NES). The intensity of the color corresponds to the statistical significance of the enrichment, derived by computing the multiple testing-adjusted permutation-based p-value using the Benjamini–Yekutieli correction. Enrichments with a statistical significance above 0.30 are represented by empty circles. Statistical significance was assessed by (**A**) paired t-test for 0 vs. 24 hr (*p<0.05) and two-way ANOVA for HS vs. TB at each time (**p<0.01); (**B**) unpaired t-test (*p<0.05; **p<0.01).

*Figure 7 continued on next page*

*Figure 7 continued*

The online version of this article includes the following figure supplement(s) for figure 7:

**Figure supplement 1.** Association between baseline glycolytic status of monocytes and the severity of lung disease.

inflammatory state. Finally, no significant enrichment of oxidative phosphorylation-associated genes was found in any of the performed comparisons (*Figure 7D*). Taken together, these results demonstrate that TB disease is associated with an increased activation and glycolytic profile of circulating CD16+ monocytes.

## HIF1A activation in CD16+ monocytes from TB patients leads to differentiated DCs with a poor migration capacity

Since circulating CD16+ monocytes from TB patients are highly glycolytic, we evaluated the expression of HIF1A among the populations. We found that CD16+ monocytes from TB patients exhibited a higher expression of HIF1A than from healthy donors (*Figure 8A*). As we previously demonstrated that CD16+ monocytes from TB patients generate aberrant DCs (*Balboa et al., 2013*), we hypothesized that the different metabolic profile of this monocyte subset could yield DCs with some sort of exhausted glycolytic capacity and thus lower migration activity upon Mtb exposure. To test this hypothesis, we treated with DMOG to increase the activity of HIF1A during the first 24 hr of monocyte differentiation from healthy donors, leading to an exacerbated increase in lactate release at early stages of the differentiation (*Figure 8B*). Such early addition of DMOG to healthy monocytes resulted in the generation of DCs (6 days with IL-4/GM-CSF) characterized by equivalent levels of CD1a as control DCs, with a significant decrease in the expression of DC-SIGN (*Figure 8—figure supplement 1A*). In terms of activation marker expression, DCs differentiated from DMOG-pretreated cells responded to iMtb by upregulating CD86 at higher levels compared to control cells, with an accompanying trend toward reduced upregulation of CD83 (*Figure 8—figure supplement 1B*). We also observed that DCs from DMOG pretreated cells exhibited a lower migratory capacity in response to iMtb (*Figure 8C*), reminiscent of the 2D migration capacities of Mo-DCs from TB patients (*Figure 8—figure supplement 2*). Altogether, our data suggest that the activated glycolytic status of monocytes from TB patients leads to the generation of DCs with low motility in response to Mtb.

## Discussion

In this study, we provide evidence for the role of HIF1A-mediated glycolysis in promoting the migratory capacity of DCs upon encounter with iMtb. Our approach to quantify the ex vivo metabolism of monocytes shows that CD16+ monocytes from TB patients display an exacerbated glycolytic activity that may result in the generation of DCs with poor migratory capacities in response to iMtb. Our results suggest that under extensive chronic inflammatory conditions, such as those found in TB patients, circulating monocytes may be metabolically preconditioned to differentiate into DCs with low migratory potential (*Figure 8—figure supplement 2*).

Upon Mtb infection of naive mice, initial accumulation of activated CD4+ T cells in the lung is delayed, occurring between 2–3 weeks post-infection (*Wolf et al., 2007*; *Reiley et al., 2008*). The absence of sterilizing immunity induced by TB vaccines, such as BCG, has been proposed to result from delayed activation of DCs and the resulting delay in antigen presentation and activation of vaccine-induced CD4+ T-cell responses (*Griffiths et al., 2016*). In this context, it was demonstrated that Mtb-infected Mo-DCs recruited to the site of infection exhibit low CCR7 expression and impaired migration to lymph nodes compared to uninfected Mo-DCs (*Harding et al., 2015*). Additionally, Mo-DCs have been found to play a key role in transporting Mtb antigens from the lung to the draining lymph node, where conventional DCs present antigens to naive T cells (*Samstein et al., 2013*). The migratory capacity of responding DCs is thus of paramount importance to the host response to Mtb infection.

Here, we found that Mtb exposure triggers glycolysis in Mo-DCs from healthy donors, which promotes their migration capacity in an HIF1A-dependent manner. Recently, it was shown that glycolysis was required for CCR7-triggered murine DC migration in response to LPS (*Guak et al., 2018*; *Liu et al., 2019*; *Everts et al., 2014*). Glycolysis was also reported to be required for the migration of

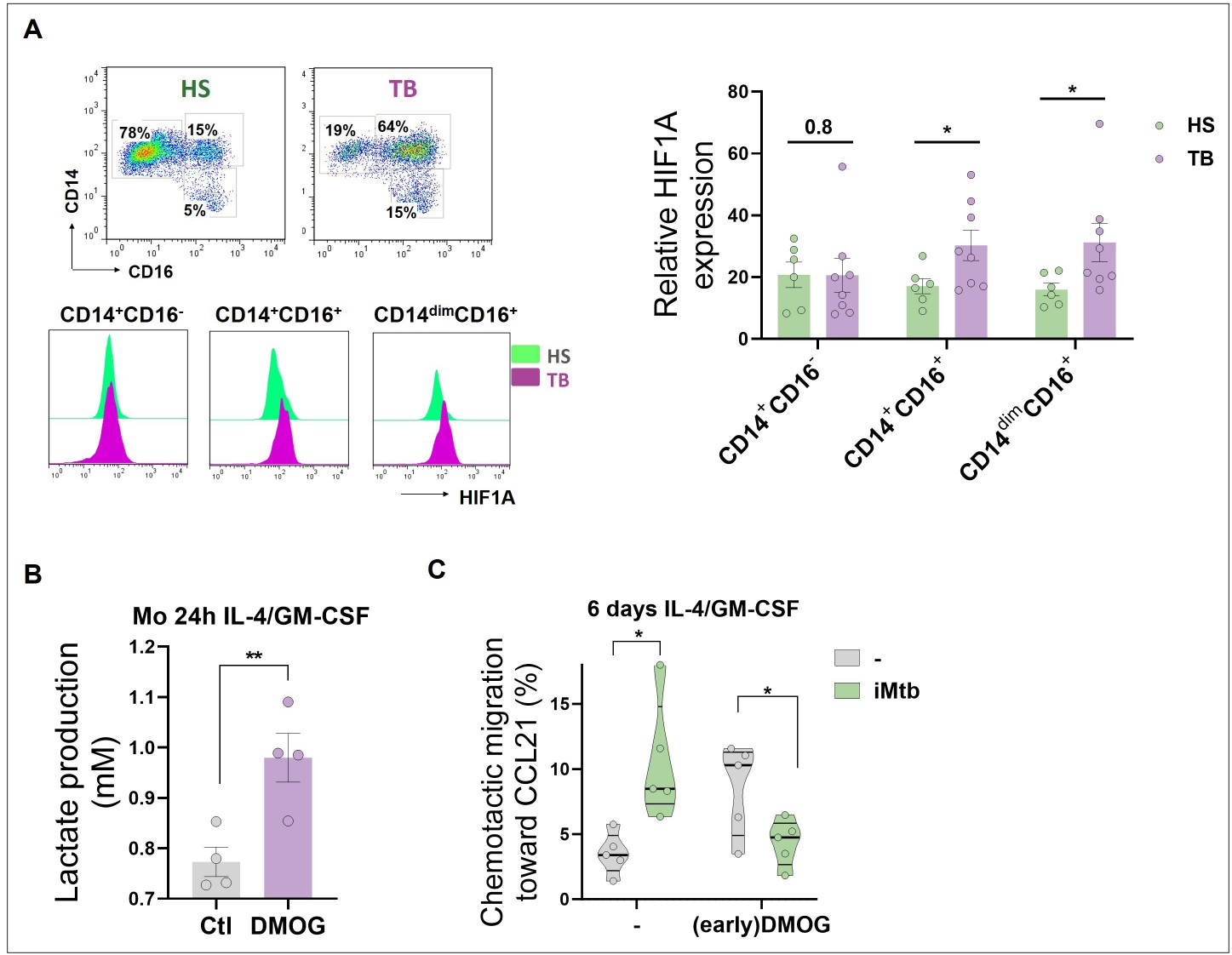

**Figure 8.** HIF1A activation in CD16+ monocytes from tuberculosis (TB) patients leads to dendritic cells (DCs) with poor migration capacity. (**A**) Ex vivo determination of HIF1A expression by monocytes from healthy subjects (HS) or TB patients (TB) for each monocyte subset (CD14+CD16-, CD14+CD16+, and CD14dimCD16+) (N = 6). (**B, C**) Monocytes from HS were treated with dimethyloxalylglycine (DMOG) during the first 24 hr of differentiation with IL-4/GM-CSF (earlyDMOG) and removed afterward. On day 6 of differentiation, cells were stimulated (or not) with irradiated *Mycobacterium tuberculosis* (iMtb). (**B**) Monocyte lactate release after 24 hr of DMOG addition (N = 4). (**C**) Chemotactic activity toward CCL21 of DCs (N = 5). Statistical significance was assessed by (**B**) paired *t*-test (p<0.01); (**C**) two-way ANOVA followed by Tukey's multiple-comparisons test (*p<0.05). The data are represented as scatter plots, with each circle representing a single individual, means ± SEM are shown.

The online version of this article includes the following figure supplement(s) for figure 8:

**Figure supplement 1.** Impact of premature activation of HIF1A in monocytes on the generated dendritic cells (DCs).

**Figure supplement 2.** Dual role of the glycolysis/HIF1A axis on dendritic cell (DC) migration in tuberculosis (TB).

other immune cells such as macrophages (*Semba et al., 2016*) and regulatory T cells (*Kishore et al., 2017*). Consistently, we show that inhibition of HIF1A-dependent glycolysis impairs human Mo-DC migration upon Mtb stimulation. The link between cellular metabolism and migratory behavior is supported by studies that have elucidated how glycolysis can be mechanically regulated by changes in the architecture of the cytoskeleton, ultimately impacting the activity of glycolytic enzymes (*Park et al., 2020*; *Fernie et al., 2020*). In addition, interesting links between cellular mechanics and metabolism have been previously described for DCs, highlighting the potential to alter DC mechanics to control DC trafficking and consequently T cell priming (*Currivan et al., 2022*). However, studies

focused on the molecular mechanisms by which metabolic pathways impact the machinery responsible for cell movement in the context of TB infection will be required to better understand and design therapeutic manipulation.

Our research indicates that DCs exhibit upregulated glycolysis following stimulation or infection by Mtb. This metabolic shift is crucial for facilitating cell migration to the draining lymph nodes, an essential step in mounting an effective immune response. Yet, it remains uncertain whether this glycolytic induction reaches a threshold conducive to generating a protective immune response, a matter that our findings do not definitively address. In addition, we demonstrated that tolerogenic DCs induced by DX as well as DCs derived from TB patient monocytes exhibit lower lactate release and impaired trafficking toward CCL21 upon Mtb stimulation; both phenotypes could be rescued by stabilization of HIF1A expression. To our knowledge, this is the first study to address how the metabolic status of monocytes from TB patients influences the migratory activity of further differentiated DCs. According to our findings, the activation status of the glycolysis/HIF1A axis in monocytes would be a predictor of refractoriness to differentiation into migratory DCs in TB. With respect to the metabolism of tolerogenic DCs broadly, our results are consistent with reported data showing that DC tolerance can be induced by drugs promoting OXPHOS, such as vitamin D and DX (*Ferreira et al., 2012*; *Ferreira et al., 2009*; *Basit and de Vries, 2019*). It was interesting to note that, although migration of tolerogenic DCs did not increase upon Mtb stimulation, it was increased under basal conditions, which agrees with previous data showing a high steady-state migration capacity of putatively tolerogenic DCs (*Ohl et al., 2004*).

It has been widely demonstrated that immune cells can switch to glycolysis following engagement of TLRs (*Krawczyk et al., 2010*). Our work showed that TLR2 ligation by either viable or irradiated Mtb was necessary to trigger glycolysis in DCs, at least at early times post-stimulation. In fact, even bystander DCs increased their glycolytic activity in Mtb-infected cultures, suggesting that mycobacterial antigens or bacterial debris present in the microenvironment may be sufficient to trigger TLR-dependent glycolysis. In the context of natural infection in vivo, we foresee that DC with different levels of infection will coexist, some with low bacillary load that, according to our data, may be able to trigger glycolysis and migrate, while others highly infected DCs would more likely die (*Ryan et al., 2011*). It remains to be elucidated whether persistent interaction between DCs and Mtb might lead to an attenuation in glycolysis over time, as has been reported for macrophages (*Hackett et al., 2020*). In this regard, our data demonstrates that chronic Mtb infection leads to monocytes bearing an exacerbated glycolytic status likely tied to prolonged and/or excessive stimulation of membrane-bound TLRs in circulation, which results in DCs with an exhausted glycolytic capacity. Although DCs stimulated with iMtb in the presence of an HIF1A inhibitor exhibited differences in activation markers and cytokine profile, we found that they were still able to activate CD4[+] T cells from PPD+ donors in response to iMtb. These findings complement previous evidence showing that LPS-induced mature DCs inhibit T-cell responses through HIF1A activation in the presence of glucose, leading to greater T cell activation capacity in low glucose contexts such as at the interface between DCs and T cells (*Lawless et al., 2017*). In this work, we did not detect an impact on T cell activation upon HIF1A inhibition in DCs, but we observed a clear reduction in their migration capacity that may limit or delay DC encounters with T cells in vivo, leading to poor T cell activation in the lymph nodes. In this regard, mouse studies have shown that DC migration directly correlates with T cell proliferation (*Martín-Fontecha et al., 2003*). However, we cannot rule out the possibility that other CD4[+] T cell subsets (such as regulatory T cells), CD1-restricted T cells, and/or CD8[+] T cell subsets could be differentially activated by iMtb-stimulated DCs lacking HIF1A activity.

Three different populations of human monocytes have been identified: classical (CD14[+], CD16[−]), intermediate (CD14[+], CD16[+]), and non-classical (CD14[dim], CD16[+]) monocytes (*Ziegler-Heitbrock et al., 2010*). These monocyte subsets are phenotypically and functionally distinct. Classical monocytes readily extravasate into tissues in response to inflammation, where they can differentiate into macrophage-like or DC-like cells *Ginhoux and Jung, 2014*; intermediate monocytes are well-suited for antigen presentation, cytokine secretion, and differentiation; and non-classical monocytes are involved in complement and Fc gamma-mediated phagocytosis and their main function is cell adhesion (*Wong et al., 2011*; *Cros et al., 2010*). Unlike non-classical monocytes, the two CD14[+] monocyte populations are known to extravasate into tissues and thus are likely to act as precursors capable of giving rise to Mo-DCs in inflamed tissues. However, the DC differentiation capacity of the intermediate population

is still not well defined. We previously demonstrated that monocytes from TB patients generate aberrant DCs, and that CD16+ monocytes generate aberrant DCs upon treatment with GM-CSF and IL-4 (*Cougoule et al., 2018*). Here, we demonstrated that glycolysis seems to play a dual role during DC differentiation from monocytes, on the one side, being required for fully differentiated-DC migration to lymph nodes in response to Mtb and, on the other side, leading to DCs with poor iMtb-responsive migratory capacity if activated during the onset of DC differentiation (*Figure 8—figure supplement 2*). In this regard, DCs from healthy subjects respond to iMtb by inducing a glycolytic and migratory profile, while monocytes isolated from TB patients exhibit an unusual early glycolytic state that results in the ulterior generation of DCs with low glycolytic and migratory activities in response to Mtb. Similarly, we found that CD16+ cells from TB patients display an activated glycolytic status, as well as elevated HIF1A expression levels compared to their healthy counterparts. Additionally, we showed that monocytes from TB patients are not only enriched in CD16+ cells, but also display an altered chemokine receptor expression profile (*Balboa et al., 2011*), demonstrating that both phenotype and function of a given monocyte subset may differ under pathological conditions. While it is difficult to determine whether the heightened glycolytic profile of monocytes may limit their differentiation into DCs in vivo, we provided evidence that an increase in HIF1A-mediated glycolysis in precursors leads to the generation of cells with poor ability to migrate in response to CCL21 in vitro. In line with this observation, a recent study revealed a significant increase in the glycolytic capacity occurs during the first 24 hr of monocyte differentiation towards a tolerogenic DC phenotype, as induced by vitamin D3 (*Everts and Pearce, 2014*), highlighting the detrimental role of an early activated inflammatory profile in DC precursors. A possible explanation for these effects may be found in lactate accumulation in monocytes during DC differentiation. Lactate signaling in immune cells leads to metabolic alterations in DCs that program them to a regulatory state (*Manoharan et al., 2021*), and lactate has also been shown to suppress DC differentiation and maturation (*Wculek et al., 2019*); thus, excessive precursor glycolytic activity may result in DCs biased toward regulatory functions.

Taken together, our data offer new insights into the immunometabolic pathways involved in the trafficking of DCs to the lymph nodes. These insights may have various implications depending on factors such as timing, cell type, and location induction of the HIF1A/glycolysis axis. On the one hand, nurturing HIF1A-mediated glycolytic activity in DCs during the early stages of infection could potentially enhance the effectiveness of preventive strategies for TB. Particularly noteworthy is the significant impact revealed in studies where the number of DCs reaching the lymph node proved to be a crucial factor in determining the success of DC-based vaccination (*MartIn-Fontecha et al., 2003*). On the other hand, premature activation of glycolysis in precursors, as observed in CD16+ monocytes from severe TB patients, could disrupt the delicate balance necessary for an optimal immune response. This variability is consistent with the paradigm of 'too much, too little', as demonstrated by the dual roles of IFN-γ (*Kumar, 2017*) and TNF-α (*Mootoo et al., 2009*) in the context of TB. It also underscores the vital importance of maintaining an equilibrium in inflammatory responses. This study lays the foundation for further exploration into the potential systemic impact of the HIF1A/glycolysis axis within the realm of chronic inflammation contrasting with its role in a local setting during the acute phase of infection. By enhancing our understanding, these findings aim to guide the development of innovative preventive and therapeutic strategies for TB.

# Materials and methods

**Key resources table**

| Reagent type (species) or resource | Designation | Source or reference | Identifiers | Additional information |
|---|---|---|---|---|
| Antibody | Anti-human TLR2 | BioLegend | Cat# 309717 | |
| Antibody | Anti-human TLR4 | BioLegend | Cat# 312813 | |
| Antibody | Anti-human CD1a | eBioscience | RRID:AB_467039 | |
| Antibody | Anti-human DC-SIGN | R&D System | Cat# MAB161 | |
| Antibody | Anti-human CD14 | BD Biosciences | Cat# 557154 | |
| Antibody | Anti-human CD86 | BioLegend | Cat# 374216 | |

*Continued on next page*

*Continued*

| Reagent type (species) or resource | Designation | Source or reference | Identifiers | Additional information |
|---|---|---|---|---|
| Antibody | Anti-human CD83 | eBioscience | Cat# 14-0839-82 | |
| Antibody | Anti-human CD274 (B7-H1, PD-L1) | BD Pharmingen | Cat# 557924 | |
| Antibody | Anti-human Glut1 | R&D System | Cat# MAB1418 | |
| Antibody | Anti-human HIF1A | BioLegend | Cat# 359704 | |
| Antibody | Anti-human CD4 | BioLegend | Cat# 357402 | |
| Antibody | Anti-human CXCR3 | BioLegend | Cat# 353719 | |
| Antibody | Anti-human CCR4 | BD Biosciences | Cat# 560726 | |
| Antibody | Anti-human CCR6 | BD Biosciences | Cat# 560619 | |
| Antibody | Anti-human CD3 | BioLegend | Cat# 300317 | |
| Antibody | Anti-human CD16 | BioLegend | Cat# 302008 | |
| Antibody | Anti-mouse CD11c | BD Pharmingen | Cat# 561044 | |
| Biological sample (*Mycobacterium tuberculosis*) | *M. tuberculosis* H37Rv | N/A | N/A | |
| Biological sample (*M. tuberculosis*) | Tuberculosis γ-irradiated H37Rv | BEI Resource | Cat# NR-49098 | |
| Biological sample | Patients-derived blood | Hospital F.J.Muñiz (Buenos Aires, Argentina) | N/A | |
| Biological sample | Buffy coats from healthy donors | Centro Regional de Hemoterapia Garrahan (Buenos Aires, Argentina) | N/A | |
| Biological sample | Blood from PPD+ healthy donors | N/A | N/A | |
| Peptide, recombinant protein | Recombinant human GM-CSF | Peprotech | Cat# 300-03 | |
| Peptide, recombinant protein | Recombinant human IL-4 | BioLegend | Cat# 430307 | |
| Peptide, recombinant protein | Recombinant mouse GM-CSF | BioLegend | Cat# 576304 | |
| Peptide, recombinant protein | Recombinant mouse IL-4 | BioLegend | Cat# 574302 | |
| Chemical compound, drug | Lipopolysaccharides from *Escherichia coli* O127:B8 | Sigma-Aldrich | Cat# 93572-42-0 | |
| Chemical compound, drug | Dexamethasone | Sidus | Cat# 229197-1 | |
| Chemical compound, drug | PX-478 2HCl | Selleck Chemicals | Cat# S7612 | |
| Chemical compound, drug | DMOG | Bertin Technologies | Cat# 300-02 | |
| Chemical compound, drug | GSK2837808A | Cayman Chemical | Cat# 1445879-21-9 | |
| Chemical compound, drug | Echinomycin | Cayman Chemical | Cat# 512-64-1 | |
| Chemical compound, drug | Sodium oxamate | Cayman Chemical | Cat# 565-73-1 | |
| Peptide, recombinant protein | Recombinant Human Exodus-2 (CCL21) | Peprotech | Cat# 300-35A | |
| Chemical compound, drug | Collagen from calf skin | Sigma-Aldrich | Cat# C9791-10MG | |
| Commercial assay or kit | Lactate Kit | Wiener | Cat# 1999795 | |
| Commercial assay or kit | Glicemia Enzimática AA Kit | Wiener | Cat# 1009803 | |
| Commercial assay or kit | Perm2 solution | BD Biosciences | Cat# 340973 | |
| Commercial assay or kit | Trizol reagent | Thermo Fisher Scientific | Cat# 15596026 | |
| Commercial assay or kit | MitoSpy Green FM | BioLegend | Cat# 424805 | |
| Commercial assay or kit | TNF alpha Human ELISA Kit | eBiosciences | Cat# BMS223-4 | |

*Continued*

| Reagent type (species) or resource | Designation | Source or reference | Identifiers | Additional information |
|---|---|---|---|---|
| Commercial assay or kit | IL-10 Human ELISA Kit | eBiosciences | Cat# BMS215-2 | |
| Commercial assay or kit | IL-17A Human ELISA Kit | eBiosciences | Cat# 88-7176-22 | |
| Commercial assay or kit | IFN gamma Human ELISA Kit | eBiosciences | Cat# BMS228 | |
| Commercial assay or kit | Zombie Violet Fixable Viability Kit | BioLegend | Cat# 423113 | |
| Commercial assay or kit | SCENITH | Gifted by Rafael Argüello | N/A | |
| Sequence-based reagent | Primer: *EEF1A1* Fwd: TCGGGCAAGTCCACCACTAC | *Marín Franco et al., 2020* | N/A | |
| Sequence-based reagent | Primer: *EEF1A1* Rev: CCAAGACCCAGGCATACTTGA | *Marín Franco et al., 2020* | N/A | |
| Sequence-based reagent | Primer: *HIF1A* Fwd: ACTAGCCGAGGAAGAACTATGAA | *Marín Franco et al., 2020* | N/A | |
| Sequence-based reagent | Primer: *HIF1A* Rev: TACCCACACTGAGGTTGGTTA | *Marín Franco et al., 2020* | N/A | |
| Sequence-based reagent | Primer: *LDHA* Fwd: TGGGAGTTCACCCATTAAGC | *Marín Franco et al., 2020* | N/A | |
| Sequence-based reagent | Primer: *LDHA* Rev: AGCACTCTCAACCACCTGCT | *Marín Franco et al., 2020* | N/A | |
| Software, algorithm | ImageJ | ImageJ | https://imagej.nih.gov/ij/ | |
| Software, algorithm | Prism (v5) | GraphPad | https://www.graphpad.com/ | |
| Software, algorithm | FlowJo 7.6.5 | TreeStar | https://www.flowjo.com/ | |
| Software, algorithm | FCS Express V3 | DeNovo Software | https://www.denovosoftware.com/ | |
| Software, algorithm | Seahorse Wave | Agilent | https://www.agilent.com/ | |
| Software, algorithm | CFX Maestro | Bio-Rad | https://www.bio-rad.com/ | |
| Software, algorithm | Metamorph | Molecular Devices | https://www.moleculardevices.com/ | |

## Chemical reagents

LPS from *Escherichia coli* O111:B4 was obtained from Sigma-Aldrich (St. Louis, MO). Dexamethasone (Dx) was from Sidus (Buenos Aires, Argentina). PX-478 2HCL was purchased from Selleck Chemicals (Houston, USA) and DMOG from Santa Cruz, Biotechnology (Palo Alto, CA). Additionally, GSK2837808A was purchased from Cayman Chemical (Michigan, USA), together with echinomycin and sodium oxamate.

## Bacterial strain and antigens

Mtb H37Rv strain was grown at 37°C in Middlebrook 7H9 medium supplemented with 10% albumin-dextrose-catalase (both from Becton Dickinson, NJ) and 0.05% Tween-80 (Sigma-Aldrich). The Mtb γ-irradiated H37Rv strain (NR-49098) was obtained from BEI Resource (NIAID, NIH, USA). The RFP-expressing Mtb strain was gently provided by Dr. Fabiana Bigi (INTA, Castelar, Argentina).

## Preparation of monocyte-derived DCs

Buffy coats from healthy donors were prepared at Centro Regional de Hemoterapia Garrahan (Buenos Aires, Argentina) according to institutional guidelines (resolution number CEIANM-664/07). Informed consent was obtained from each donor before blood collection. Monocytes were purified by centrifugation on a discontinuous Percoll gradient (Amersham, Little Chalfont, UK) as previously described (*Genoula et al., 2018*). Then, monocytes were allowed to adhere to 24-well plates at $5 \times 10^5$ cells/well for 1 hr at 37°C in warm RPMI-1640 medium (Thermo Fisher Scientific, Waltham, MA). The mean purity of adherent monocytes was 85% (range: 80–92%). The medium was then supplemented to a final concentration of 10% fetal bovine serum (FBS, Sigma-Aldrich), human recombinant granulocyte-macrophage colony-stimulating factor (10 ng/ml, GM-CSF, Peprotech, NJ), and IL-4 (20 ng/ml, BioLegend, San Diego, USA). Cells were allowed to differentiate for 5–7 days (DC-SIGN+ cells in the culture >90%).

## DC stimulation

DCs were stimulated with either iMtb or viable Mtb at equivalent $OD_{600}$ doses for 24 hr at 37°C. The cells were washed three times, and their phenotype and functionality were evaluated together with survival of activated cells; cell number and viability were determined by either trypan blue exclusion assays or MTT. Infections were performed in the biosafety level 3 (BSL-3) laboratory at the Unidad Operativa Centro de Contención Biológica (UOCCB), ANLIS-MALBRAN (Buenos Aires), according to the biosafety institutional guidelines.

## DC treatments

When indicated, neutralizing monoclonal antibodies (mAb), or their corresponding isotype antibodies as mock controls, were added 30 min prior to DC stimulation to inhibit TLR2 (309717, BioLegend) or TLR4 (312813, BioLegend). In addition, DCs were incubated with PX-478 (20 µM) or echinomycin (1 nM) with the purpose of inhibiting HIF1A activity, DMOG (50 µM) to stabilize HIF1A, and oxamate (20 mM) or GSK2837808A (20 µM) to inhibit LDH. DC stimulation with iMtb occurred 30 min after treatment without drug washout.

In *Figure 6* and *Figure 6—figure supplement 1*, Dx-induced tolerogenic dendritic cells (Dx-DC) were generated by incubating DCs with 0.1 µM of Dx for 1 hr. Thereafter, cells were washed, and 'complete medium' was added. Tolerogenic Dx-DCs were then stimulated (or not) with iMtb in the presence or not of DMOG (50 µM).

## Determination of metabolite concentrations

Lactate production and glucose concentrations in the culture medium was measured using the spectrophotometric assays Lactate Kit and Glicemia Enzimática AA Kit both from Wiener (Argentina), which are based on the oxidation of lactate or glucose, respectively, and the subsequent production of hydrogen peroxide (*Barham and Trinder, 1972*). The consumption of glucose was determined by assessing the reduction in glucose levels in culture supernatants in comparison with RPMI 10% FBS. The absorbance was read using a Biochrom Asys UVM 340 Microplate Reader microplate reader and software.

## Quantitative RT-PCR

Total RNA was extracted with Trizol reagent (Thermo Fisher Scientific) and cDNA was reverse transcribed using the Moloney murine leukemia virus reverse transcriptase and random hexamer oligonucleotides for priming (Life Technologies, CA). The expression of the genes *HIF1A* and *LDHA* was determined using the PCR SYBR Green sequence detection system (Eurogentec, Seraing, Belgium) and the CFX Connect Real-Time PCR Detection System (Bio-Rad, CA). Gene transcript numbers were standardized and adjusted relative to eukaryotic translation elongation factor 1 alpha 1 (*EEF1A1*) transcripts. Gene expression was quantified using the ΔΔCt method.

## Immunofluorescence analysis

FITC-, PE-, or PerCP.Cy5.5-labeled mAbs were used for phenotypic analysis of the following cell-surface receptor repertoires: FITC-anti-CD1a (clone HI149, eBioscience), PE-anti-DC-SIGN (clone 120507, R&D System), PerCP.Cy5.5-anti-CD86 (clone 374216, BioLegend), FITC-anti-CD83 (clone HB15e, eBioscience), PE-anti-PD-L1 (clone MIH1, BD Pharmingen), and in parallel, with the corresponding isotype control antibody. Approximately $5 \times 10^5$ cells were seeded into tubes and washed once with PBS. Cells were stained for 30 min at 4°C and washed twice. Additionally, cells were stained for 40 min at 4°C with fluorophore-conjugated antibodies PE-anti-Glut1 (clone 202915, R&D Systems, MN) and in parallel, with the corresponding isotype control antibody. For HIF1A determination, DCs were permeabilized with methanol and incubated with PE-anti-HIF1A (clone 546-16, BioLegend). Stained populations were gated according to forward scatter (FSC) and side scatter (SSC) analyzed on FACScan (Becton Dickinson). Isotype matched controls were used to determine autofluorescence and nonspecific staining. Analysis was performed using the FCS Express (De Novo Software), and results were expressed as median fluorescence intensity (MFI) or percentage of positive cells.

### Soluble cytokines determinations

Supernatants from DC populations or DC-T cell cocultures were harvested and assessment of TNF-α, IL-10, IL-17A, or IFN-γ production was measured by ELISA, according to manufacturer's instructions (eBioscience). The detection limit was 3 pg/ml for TNF-α and IL-17A, 6 pg/ml for IFN-γ, and 8 pg/ml for IL-10.

### CD4+ T cell activation assay

Specific lymphocyte activation (recall) assays were carried out in cells from tuberculin purified protein derivative-positive skin test (PPD+) healthy donors by culturing DC populations and autologous T cells at a ratio of 10T cells to 1 DC in round bottom 96-well culture plates for 5 days as detailed previously (*Balboa et al., 2016*). The numbers of DCs were adjusted to live cells before the start of the co-cultures. After 5 days, CD4+ T cell subsets were identified by immunolabeling according to the differential expression of CCR4, CXCR3, and CCR6 as previously reported (*Acosta-Rodriguez et al., 2007*). CXCR3+CCR4−CCR6− (Th1), CXCR3−CCR4+CCR6− (Th2), CXCR3−CCR4+CCR6+ (Th17), and CXCR3+CCR4−CCR6+ (Th1* or Th1/Th17). The fluorochrome-conjugated antibodies used for flow cytometry analysis were CD4-FITC (clone A161A1, BioLegend), CXCR3-PE-Cy7 (clone G025H7, BioLegend), CCR4-PerCPCy5.5 (clone 1G1, BD Bioscience), CCR6-APC (clone 11A9, BD Bioscience), and CD3-APC-Cy7 (clone HIT3a, BioLegend). A viability dye, Zombie Violet (BioLegend), was used to exclude dead cells. Fluorescence Minus One (FMO) control was used to set proper gating for CXCR3-PE-Cy7, CCR4-PerCPCy5.5, and CCR6-APC detection. Cells were analyzed by fluorescence-activated cell sorting (FACS), using the BD FACSCANTO cytometer and FlowJo Software (BD Life Sciences).

### Chemotactic activity of DCs

Each DC population ($4 \times 10^5$ cells in 75 µl) was placed on the upper chamber of a transwell insert (5 µm pore size, 96-well plate; Corning), and 230 µl of media (RPMI with 0.5% FCS) with human recombinant CCL21 (200 ng/ml) (Peprotech) was placed in the lower chamber. After 3 hr, cells that had migrated to the lower chamber were removed and analyzed. The relative number of cells migrating was determined on a flow cytometer using Calibrite beads (BD Biosciences), where a fixed number of beads was included in each sample and the number of cells per 1000 beads was evaluated. Data were normalized to the number of initial cells.

### In vivo migration assay

DCs were differentiated from bone marrow precursors obtained from female 8-week naïve BALB/c mice in the presence of murine GM-CSF (10 ng/ml) and IL-4 (10 ng/ml) both from BioLegend for 7 days. After differentiation, DCs were treated with oxamate (20 mM) or PX-478 (10 uM) and stimulated with iMtb. After 24 hr, DCs were stained with CFSE (5 µM) and inoculated intradermally in the inguinal zone of naïve female 8-week BALB/c mice. Three hours post-injection, inguinal lymph nodes close to the site of inoculation were harvested and cells were stained with fluorophore-conjugated antibody PE-anti-CD11c (clone HL3, BD Pharmingen). Analysis was performed using the FlowJo Software, and results were expressed as the percentage of CFSE+/CD11c+ cells.

### 3D migration assay

$0.5 \times 10^5$ DCs were seeded on top of fibrillar collagen matrices polymerized from Nutragen 2 mg/ml, 10% v/v MEM 10X (MEM invitrogen, Carlsbad, CA), UltraPure distilled water and 4–6% v/v bicarbonate buffer (pH = 9) 7.5%. After 24 hr, cellular migration was quantified by taking images using an inverted microscope (Leica DMIRB, Leica Microsystems, Deerfield, IL) and the software Metamorph, as described previously (*Van Goethem et al., 2010*). Alternatively, in a similar manner, matrices were polymerized using Collagen (Sigma-Aldrich, C9791-10MG) in *Figure 5C and D*. After 24 hr of cellular migration, matrices were fixed with paraformaldehyde (PFA) 4% during 30 min at room temperature and stained with DAPI (Cell Signaling). Collagen was removed and membranes were mounted with DAKO. Images were taken using confocal microscopy (FluoView FV 1000), and cells were counted per field.

## Measurement of cell respiration with Seahorse flux analyzer

Bioenergetics were determined using a Seahorse XFe24 analyzer. ATP production rates and relative contribution from the glycolysis and the OXPHOS were measured by the Seahorse XF Real-Time ATP Rate Assay kit. DCs ($2 \times 10^5$ cells/well) were cultured in 3 wells per condition. The assay was performed in XF Assay Modified DMEM. Three consecutive measurements were performed under basal conditions and after the sequential addition of oligomycin and rotenone/antimycin (Agilent, USA). ECAR and OCR were measured. Mitochondrial ATP production rate was determined by the decrease in the OCR after oligomycin addition. On the other hand, the complete inhibition of mitochondrial respiration with rotenone plus antimycin A allows accounting for mitochondrial-associated acidification, and when combined with PER data, allows calculation of glycolysis ATP production rate. All OCR and ECAR values were normalized. Briefly, before the assay, brightfield imaging was performed. Cellular area per condition was calculated using ImageJ software and imported into Wave (Agilent) using the normalization function.

## SCENITH assay

SCENITH experiments were performed as previously described (**Argüello et al., 2020**) using the SCENITH kit containing all reagents and anti-puromycin antibodies (https://www.scenith.com/). Briefly, DCs or PBMCs were treated for 40 min at 37°C in the presence of the indicated inhibitors of various metabolic pathways and puromycin. After the incubation, puromycin was stained using a fluorescently labeled anti-puromycin monoclonal antibody (clone R4743L-E8) with Alexa Fluor 647 or Alexa Fluor 488, and analyzed by flow cytometry. For metabolic analysis of monocyte subsets, PBMCs were labeled with PE-anti-CD16 (clone 3G8, BioLegend) and PECy7-anti-CD14 (clone HCD14, BioLegend) mAbs. The impact of the various metabolic inhibitors was quantitated as described (**Argüello et al., 2020**).

## Transmission electron microscopy

DCs were fixed in 2.5% glutaraldehyde/2% PFA (EMS, Delta-Microscopies) dissolved in 0.1 M Sorensen buffer (pH 7.2) for 1 hr at room temperature, and then preserved in 1% PFA dissolved in Sorensen buffer. Adherent cells were treated for 1 hr with 1% aqueous uranyl acetate then dehydrated in a graded ethanol series and embedded in Epon. Sections were cut on a Leica Ultracut microtome and ultrathin sections were mounted on 200 mesh onto Formvar carbon-coated copper grids. Finally, thin sections were stained with 1% uranyl acetate and lead citrate and examined with a transmission electron microscope (Jeol JEM-1400) at 80 kV. Images were acquired using a digital camera (Gatan Orius). For mitochondrial morphometric analysis, TEM images were quantified with the ImageJ 'analyze particles' plugin in thresholded images, with size ($\mu m^2$) settings from 0.001 to infinite. For quantification, 8–10 cells of random fields (1000× magnification) per condition were analyzed.

## Changes of mitochondrial mass

Mitochondrial mass was determined in DCs by fixing the cells with PFA 4% and labeling them with the probe MitoSpy Green FM (BioLegend). Green fluorescence was analyzed by flow cytometry (FACScan, BD Biosciences).

## GSEA of human monocytes

BubbleMap analysis was performed with 1000 geneset-based permutations, and with 'Signal2Noise' as a metric for ranking the genes. The results are displayed as a BubbleMap, where each bubble is a GSEA result and summarizes the information from the corresponding enrichment plot. The color of the Bubble corresponds to the population from the pairwise comparison in which the geneset is enriched. The bubble area is proportional to the GSEA normalized enrichment score (NES). The intensity of the color corresponds to the statistical significance of the enrichment, derived by computing the multiple testing-adjusted permutation-based p-value using the Benjamini–Yekutieli correction. Enrichments with a statistical significance above 0.30 are represented by empty circles.

## Patient blood donors

TB patients were diagnosed at the División Tisioneumonología, Hospital F.J.Muñiz (Buenos Aires, Argentina) by the presence of recent clinical respiratory symptoms, abnormal chest radiography,

**Table 1.** Demographic and clinical characteristics of tuberculosis (TB) patients.

| | |
|---|---|
| Age, years (range) | **36 (19–67)** |
| Gender, % (number/total) | M, 81% (31/38)<br>F, 19% (7/38) |
| Nationality, % (number/total) | Argentina, 76.31% (29/38)<br>Bolivia, 15.78% (6/38)<br>Paraguay, 2.63% (1/38)<br>Peru, 5.26% (2/38) |
| TB disease localization, % (number/total) | Pulmonary, 94% (36/38)<br>Pulmonary + extrapulmonary, 6% (2/38) |
| AFB* in sputum, % (number/total) | 3+, 21% (8/38)<br>2+, 13% (5/38)<br>1+, 52% (20/38)<br>-, 13% (5/38) |
| Leukocyte count, mean ± SEM, cell/μl | 8483 ± 509 |
| Lymphocyte mean ± SEM, % | 19 ± 2 |
| Monocyte mean ± SEM, % | 7 ± 0.5 |

*Acid-fast-bacilli (AFB) in sputum: -, 1+, 2+, 3+ are defined according to the International Union Against Tuberculosis and Lung Disease (IUATLD)/World Health Organization (WHO) quantification scale.

and positive culture of sputum or positive sputum smear test for acid-fast-bacilli. Written, informed consent was obtained according to the Ethics Committee from the Hospital Institutional Ethics Review Committee. Exclusion criteria included HIV-positive patients and the presence of concurrent infectious diseases or comorbidities. Blood samples were collected during the first 15 days after commencement of treatment. All tuberculous patients had pulmonary TB (*Table 1*). The term symptoms evolution refers to the time period during which a patient experiences cough and phlegm for more than 2–3 weeks, with (or without) sputum that may (or not) be bloody, accompanied by symptoms of constitutional illness (e.g., loss of appetite, weight loss, night sweats, and general malaise).

## Statistics

All values are presented as the median ± SEM of 3–13 independent experiments. Each independent experiment corresponds to one donor. Each assay, which included human-derived DCs, was performed with a number of donors specified in each figure legend at a rate of two donors per time. For Seahorse assays, OCR and PER values are shown as mean ± SD. Comparisons between unpaired experimental conditions were made using either ANOVA for parametric data or Friedman test for nonparametric data followed by Dunn's multiple-comparison test. Comparisons between paired experimental conditions were made using the two-tailed Wilcoxon signed-rank test for nonparametric data or *t*-test for parametric data. Correlation analyses were determined using the Spearman's rank test. A p-value of 0.05 was considered significant.

## Study approval

### Human specimens

The study design was reviewed and approved by the Ethics Committees of the Academia Nacional de Medicina (49/20/CEIANM) and the Muñiz Hospital, Buenos Aires, Argentina (NI #1346/21). All participants voluntarily enrolled in the study by signing an informed consent form after receiving detailed information about the research study.

### Mouse studies

All experimental protocols were approved by the Institutional Animal Care and Use of the Experimentation Animals Committee (CICUAL number 090/2021) of the Institute of Experimental Medicine (IMEX, Buenos Aires).

## Acknowledgements

We thank the staff of the Regional Center of Hemotherapy of the Garrahan Hospital (Buenos Aires). We greatly thank Claire Lastrucci for designing the model illustration. In addition, we are grateful for the editing service provided by Life Science editors. This work was supported by the Argentinean National Agency of Promotion of Science and Technology (PICT-2019-01044 and PICT-2020-00501 to LB); the Argentinean National Council of Scientific and Technical Investigations (CONICET, PIP 11220200100299CO to LB); the Centre National de la Recherche Scientifique, *Université Paul Sabatier*, the *Agence Nationale de Recherche sur le Sida et les hépatites virales (ANRS)* (ANRS2018-02, ECTZ 118551/118554, ECTZ 205320/305352, ANRS ECTZ103104 and ECTZ101971 to CV, ON, and GL-V); and the French ANR JCJC-Epic-SCENITH ANR-20-CE14-0028 and CoPoC Inserm-transfert MAT-PI-17493-A-04 to RA. The funders had no role in study design, data collection, and analysis, decision to publish, or preparation of the manuscript.

## Additional information

### Funding

| Funder | Grant reference number | Author |
|---|---|---|
| Agencia Nacional de Promoción de la Investigación, el Desarrollo Tecnológico y la Innovación | PICT-2019-01044 | Luciana Balboa |
| Agencia Nacional de Promoción Científica y Tecnológica | PICT-2020-00501 | Luciana Balboa |
| Consejo Nacional de Investigaciones Científicas y Técnicas | 11220200100299CO | Luciana Balboa |
| Agence Nationale de Recherches sur le Sida et les Hépatites Virales | ANRS2018-02 | Christel Vérollet |
| Agence Nationale de Recherches sur le Sida et les Hépatites Virales | ECTZ 118551/118554 | Geanncarlo Lugo Villarino Christel Vérollet |
| Agence Nationale de Recherches sur le Sida et les Hépatites Virales | ECTZ 205320/305352 | Olivier Neyrolles Christel Vérollet |
| Agence Nationale de Recherches sur le Sida et les Hépatites Virales | ECTZ103104 | Christel Vérollet |
| Agence Nationale de Recherches sur le Sida et les Hépatites Virales | ECTZ101971 | Olivier Neyrolles |
| Agence Nationale de la Recherche | ANR-20-CE14-0028 | Rafael J Argüello |
| Inserm Transfert | MAT-PI-17493-A-04 | Rafael J Argüello |
| The Argentinean National Council of Scientific and Technical Investigations | CONICET | Luciana Balboa |
| The Argentinean National Council of Scientific and Technical Investigations | PIP 11220200100299CO | Luciana Balboa |

| Funder | Grant reference number | Author |
|---|---|---|
| The Centre National de la Recherche Scientifique, Université Paul Sabatier, the Agence Nationale de Recherche sur le Sida et les hépatites virales (ANRS) | ANRS2018-02 | Christel Vérollet Olivier Neyrolles Geanncarlo Lugo Villarino |
| The Centre National de la Recherche Scientifique, Université Paul Sabatier, the Agence Nationale de Recherche sur le Sida et les hépatites virales (ANRS) | ECTZ 118551/118554 | Christel Vérollet Olivier Neyrolles Geanncarlo Lugo Villarino |
| The Centre National de la Recherche Scientifique, Université Paul Sabatier, the Agence Nationale de Recherche sur le Sida et les hépatites virales (ANRS) | ECTZ 205320/305352 | Christel Vérollet Olivier Neyrolles Geanncarlo Lugo Villarino |
| The Centre National de la Recherche Scientifique, Université Paul Sabatier, the Agence Nationale de Recherche sur le Sida et les hépatites virales (ANRS) | ANRS ECTZ103104 | Christel Vérollet Olivier Neyrolles Geanncarlo Lugo Villarino |
| The Centre National de la Recherche Scientifique, Université Paul Sabatier, the Agence Nationale de Recherche sur le Sida et les hépatites virales (ANRS) | ECTZ101971 | Christel Vérollet Olivier Neyrolles Geanncarlo Lugo Villarino |
| The French ANR JCJC-Epic-SCENITH | ANR-20-CE14-0028 | Rafael J Argüello |
| CoPoC Inserm-transfert | MAT-PI-17493-A-04 | Rafael J Argüello |

The funders had no role in study design, data collection and interpretation, or the decision to submit the work for publication.

## Author contributions

Mariano Maio, Conceptualization, Formal analysis, Investigation, Visualization, Methodology, Writing – original draft, Writing – review and editing; Joaquina Barros, Marine Joly, Zoi Vahlas, José Luis Marín Franco, Melanie Genoula, Sarah C Monard, María Belén Vecchione, Virginia Gonzalez Polo, Investigation; Federico Fuentes, Conceptualization, Investigation, Visualization, Methodology; María Florencia Quiroga, Conceptualization, Methodology; Mónica Vermeulen, Conceptualization, Investigation, Methodology; Thien-Phong Vu Manh, Conceptualization, Formal analysis, Visualization, Methodology, Writing – review and editing; Rafael J Argüello, Resources, Funding acquisition, Writing – original draft; Sandra Inwentarz, Rosa Musella, Lorena Ciallella, Pablo González Montaner, Domingo Palmero, Resources; Geanncarlo Lugo Villarino, Olivier Neyrolles, Resources, Funding acquisition, Writing – original draft, Writing – review and editing; María del Carmen Sasiain, Resources, Writing – original draft; Christel Vérollet, Conceptualization, Resources, Funding acquisition, Investigation, Visualization, Methodology, Writing – original draft, Writing – review and editing; Luciana Balboa, Conceptualization, Resources, Formal analysis, Funding acquisition, Investigation, Visualization, Methodology, Writing – original draft, Writing – review and editing

## Author ORCIDs

Sarah C Monard ⓘ https://orcid.org/0000-0001-7149-312X
Thien-Phong Vu Manh ⓘ http://orcid.org/0000-0002-0294-342X
Rafael J Argüello ⓘ http://orcid.org/0000-0001-9785-3883
Sandra Inwentarz ⓘ http://orcid.org/0000-0002-7526-1577
Christel Vérollet ⓘ https://orcid.org/0000-0002-1079-9085
Luciana Balboa ⓘ http://orcid.org/0000-0001-7045-3572

## Ethics

The study design was reviewed and approved by the Ethics Committees of the Academia Nacional de Medicina (49/20/CEIANM) and the Muñiz Hospital, Buenos Aires, Argentina (NI #1346/21). All participants voluntarily enrolled in the study by signing an informed consent form after receiving detailed information about the research study.

All experimental protocols were approved by the Institutional Animal Care and Use of the Experimentation Animals Committee (CICUAL number 090/2021) of the Institute of Experimental Medicine (IMEX, Buenos Aires).

Reviewer #2 (Public Review): https://doi.org/10.7554/eLife.89319.4.sa1

Author response https://doi.org/10.7554/eLife.89319.4.sa2

## Additional files

### Supplementary files

• MDAR checklist

### Data availability

All data generated or analysed during this study are included in the manuscript and supporting files.

The following previously published dataset was used:

| Author(s) | Year | Dataset title | Dataset URL | Database and Identifier |
|---|---|---|---|---|
| Burel J, Seumois G, Sette A, Peters B, Vijayanand P | 2023 | Transcriptomic profile of circulating monocyte subsets in active versus latent tuberculosis | https://www.ncbi.nlm.nih.gov/geo/query/acc.cgi?acc=GSE185372 | NCBI Gene Expression Omnibus, GSE185372 |

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
