## [Editor Report · eLife assessment]

This **useful** study tests the hypothesis that monocytes purified from tuberculosis patients differentiate into dendritic cells with different migratory capacities. The authors conclude that these monocytes are metabolically pre-conditioned to differentiate, with reduced expression of Hif1a and a glycolytically exhaustive phenotype, resulting in low migratory and immunologic potential. Overall, the evidence provided is **convincing**, advancing the field substantively and providing novel insights.

---

## [Referee Report · Reviewer #2 (Public Review)]

In the manuscript by Maio et al, the authors examined the bioenergetic mechanisms involved in the delayed migration of DC's during Mtb infection. The authors performed a series of in vitro infection experiments including bioenergetic experiments using the Agilent Seahorse XF, and glucose uptake and lactate production experiments. Also, data from SCENITH is included in the revised manuscript as well as some clinical data. This is a well written manuscript and addresses an important question in the TB field.

---

## [Author Response]

The following is the authors’ response to the previous reviews.

**eLife assessment**
This useful study tests the hypothesis that Mycobacterium tuberculosis infection increases glycolysis in monocytes, which alters their capacity to migrate to lymph nodes as monocyte-derived dendritic cells. The authors conclude that infected monocytes are metabolically pre-conditioned to differentiate, with reduced expression of Hif1a and a glycolytically exhaustive phenotype, resulting in low migratory and immunologic potential. However, the evidence is incomplete as the use of live and dead mycobacteria still limits the ability to draw firm conclusions. The study will be of interest to microbiologists and infectious disease scientists.

In response to the general eLife assessment, we would like to emphasize that the study did not deal with “infected monocytes” *per se* but rather with monocytes purified from patients with active TB. We show that monocytes purified from these TB patients (*versus* healthy controls) differentiate into DCs with different migratory capacities. In addition, to address the reviewer's comments in this new version of our manuscript, we include a relevant characterization of the migration capacity of DCs infected with Mtb to the plethora of assays already shown with viable bacteria in the previous revised version of our manuscript.

All in all, we believe that our study has significantly improved thanks to the feedback provided by the editor and reviewer panel during the different revision processes. We sincerely hope that this version of our manuscript is deemed fit for publication in this prestigious journal.

**Public Reviews:**

**Reviewer #3 (Public Review):**
In the revised manuscript by Maio et al, the authors examined the bioenergetic mechanisms involved in the delayed migration of DC's during Mtb infection. The authors performed a series of in vitro infection experiments including bioenergetic experiments using the Agilent Seahorse XF, and glucose uptake and lactate production experiments. Also, data from SCENITH is included in the revised manuscript as well as some clinical data. This is a well written manuscript and addresses an important question in the TB field. A remaining weakness is the use of dead (irradiated) Mtb in several of the new experiments and claims where iMtb data were used to support live Mtb data. Another notable weakness lies in the author's insistence on asserting that lactate is the ultimate product of glycolysis, rather than acknowledging a large body of historical data in support of pyruvate's role in the process. This raises a perplexing issue highlighted by the authors: if Mtb indeed upregulates glycolysis, one would expect that inhibiting glycolysis would effectively control TB. However, the reality contradicts this expectation. Lastly, the examination of the bioenergetics of cells isolated from TB patients undergoing drug therapy, rather than studying them at their baseline state is a weakness.

We thank the reviewer for this insightful assessment and feedback of our study. With regards to the data obtained with iMtb to support that with live Mtb, we have clarified the use of either iMtb or Mtb for each figure legend in the new version of the manuscript. Furthermore, we included the confirmation of the involvement of TLR2 ligation in the up-regulation of HIF-1α triggered by viable Mtb (new Fig S2E). We also conducted migration assays using (live) Mtb-infected dendritic cells (DCs) treated with either oxamate or PX-478 to validate that the HIF1a/glycolysis axis is indeed essential for DC migration (new Fig 5D).

We respectfully acknowledge the reviewer's statement regarding the potential relationship between glycolysis and the control of TB. However, we find it necessary to elaborate on our stance, as our data offer a nuanced perspective. Our research indicates that DCs exhibit upregulated glycolysis following stimulation or infection by Mtb. This metabolic shift is crucial for facilitating cell migration to the draining lymph nodes, an essential step in mounting an effective immune response. Yet, it remains uncertain whether this glycolytic induction reaches a threshold conducive to generating a protective immune response, a matter that our findings do not definitively address. This aspect is carefully discussed in the manuscript, lines 380-385.

Moreover, analyses of samples from chronic TB patients suggest that the outcome of inhibiting glycolysis may vary depending on factors such as the infection stage, the targeted cell type (e.g., monocytes, DCs), and the affected compartment (systemic *versus* local). This variability aligns with the concept of "too much, too little" exemplified by the dual roles of IFNγ (PMID: 28646367) and TNFα (PMID: 19275693) in TB, emphasizing the need to maintain an inflammatory equilibrium. In the context of the HIF1α/glycolysis axis, it appears to be a matter of timing: a case of "too early" activation of glycolysis in precursors, which could upset the delicate balance necessary for an effective immune response. We have added these comments in the discussion (pages 19-20, lines 468-485).

In summary, while acknowledging the reviewer's perspective, we believe that a comprehensive understanding of the interplay between Mtb infection and glycolysis in myeloid cells requires further consideration of various contextual conditions, urging caution against oversimplified interpretations.

With regard to the patients' information, as pointed out by the reviewer, according to the inclusion criteria for patient samples in the approved protocol by the Institutional Ethics Committee, we recruit patients who have received less than 15 days of treatment (for sensitive TB, the total treatment duration is at least 6 months). We do not have access to patient sample before they begin the treatment, as starting therapy is the most urgent matter in this case. Following the reviewer's suggestion, we investigated whether the glycolytic activity of monocytes correlated with the initiation of antibiotic treatment within this 15-day period. Our observations did not show any significant impact during the initial 15 days of treatment (see expanded reply below). However, after 2 months of treatment, we found that the glycolytic profile of CD16+ monocytes returned to baseline levels as per our analysis. This suggests that despite the normalization of glycolytic activity with antibiotic therapy, heightened basal glycolysis remains noticeable during the initial two weeks of treatment (time limit to meet the inclusion criteria in our study cohort).

**Recommendations for the authors:**

**Reviewer #3 (Recommendations For The Authors):**
(1) In the revised manuscript, the authors addressed concerns related to using irradiated Mtb, a positive development. However, the study predominantly employs 1:1 or 2:1 MOI, representing a low infection model, with no observed statistical distinction between the two MOIs (Fig-1). To enhance the study, inclusion of a higher MOI (e.g., 5:1 or 10:1) would have been more informative. This becomes crucial as prior research on human macrophages indicates that Mtb infection typically hampers glycolysis, a finding inconsistent with the present study.

As the reviewer notes, important work has documented the inhibition of glycolysis in *M. tuberculosis*-infected macrophages dependent on the MOI (PMID 30444490). For instance, in this study, hMDMs infected at an MOI of 1 showed increased extracellular acidification and glycolytic parameters, as opposed to macrophages infected at higher MOI, or the same MOI but measured in THP1 cells. In light of these findings, we attempted to extend our study with Mo-DCs to higher MOIs, but too much cell death was induced, limiting our ability to obtain reliable metabolic measurements and functional assays from these cultures. Consistent with this, other authors reported that more than 40% of Mo-DC die after 24 hours following infection with H37Rv at an MOI of 10 (PMID 22024399, Fig 2B). We acknowledge that more comprehensive focused *in vivo* studies would be needed to assess the overall impact of infection. We foresee that in the context of natural infection, DC with different levels of infection will coexist, some with low bacillary load that may be able to trigger glycolysis and migrate, others highly infected and more likely to die. In this case, we are unable to provide a full explanation for the delay in the onset of the adaptive response, an aspect that requires further investigation. From our perspective, the important contribution of our work is more focused on understanding the later stage of infection, when chronic infection is established, where precursors already seem to have a limited capacity to generate DC with a good migratory performance regardless of being confronted with a low bacillary load.

To better clarify the scope and limitations of the work, we added these comments to the discussion (see discussion, lines 405-408).

The study emphasizes that Mtb infection enhances glycolysis in Mo-DCs (Fig-1 and Fig-2). Despite the authors advocating lactate as the end product (citing three reviews/opinions), the historical literature supported by detailed experimentation convincingly favors pyruvate. While the authors' attempt to support an alternate glycolytic paradigm is understandable, it is simply not necessary. This is further supported by the authors' claim that oxamate is an inhibitor of glycolysis (abstract and main text). Oxamate is a pyruvate analogue that directly inhibits the conversion of pyruvate into lactate by lactate dehydrogenase. Simply put, if oxamate was an inhibitor of glycolysis then the cells would have died.

(2) Taking into account the reviewer's suggestions, we changed the text accordingly, referring to oxamate as an LDH inhibitor, including in the abstract.

In Fig-2, clarify the term "bystander DCs." Explain why these MtbRFP- DCs exhibit distinct behavior compared to uninfected DCs, especially considering their similarity to Mtb-infected ones.

(3) To clarify these results, as correctly suggested by the reviewer, we incorporated a sentence in the results section, stating that bystander DCs are cells that are not in direct association with Mtb (Mtb-RFP-DCs), but are rather nearby and exposed to the same environment (page 7, line 145-148). In other words, bystander cells are those exposed to the same secretome and soluble factors as infected cells. Our data indicate that bystander DCs upregulate their state of glycolysis just like infected DCs do, which suggests the presence of soluble mediators induced during infection that are capable of triggering glycolysis even in uninfected cells.

These results are in line with the observation that bacteria lacking infectious capacity (such as the irradiated Mtb) also trigger glycolysis in DCs (Fig 1), likely *via* TLR2 receptors that are potentially activated by the release of mycobacterial antigens or bacterial debris present in the microenvironment (Fig 3). We incorporated this interpretation in the discussion of the manuscript (lines 403-408).

(4) Notably, the authors conducted SCENITH on both iMtb and viable Mtb (Fig-2). However, OCR, PER, and Mito- & Glyco- ATP were solely measured in MO-DCs stimulated by iMtb. Given the distinct glycolytic responses between iMtb and viable Mtb, it is crucial to assess these parameters in Mo-DCs treated with viable Mtb. Moreover, it is unclear as to how the relative ATP in Fig-2F was calculated as both Mito-ATP and Glyco-ATP is significantly high in iMtb-treated Mo-DCs (Fig-2E). Also, figure 2 contains panels with no labeling, which is confusing.

We appreciate the reviewer's suggestion that additional determinations would enrich the bioenergetic profile of DCs during infection. However, due to biosafety considerations and economic-driven limitations, we are currently unable to measure OCR, PER, and Mito- & Glyco- ATP, as these assessments require live cell cultures within BSL3 containment, if live Mtb is to be employed. Regrettably, our BSL3 facility is not equipped with a Seahorse instrument—few facilities in the world have such type of BLS3-driven investment. For this key reason, we employed SCENITH for our BSL3-based experiments.

Concerning the how ATP was calculated, we show below the raw data for Mito-ATP and Glyco-ATP results and calculations of their relative contributions.

**Author response table 1. sa2table1:** 

	-	iMtb	-	iMtb
Exp 1	12,7	30,1	1,45	10,30
Exp 2	4,7	7,3	6,68	13,43
Exp 3	16,77	25,67	6,74	18,98
Exp 4	2,34	21,33	0,12	3,02
Exp 5	9,8	31	1,46	6,03
Exp 6	12	25	1,00	20,70

(5) In Figures 3, 4, & 5, the consistent use of only iMtb was observed. Previous concerns about this approach were raised in the review, with the authors asserting that the use of viable Mtb was beyond the manuscript's scope. However, this claim is inaccurate. Both the authors' findings and literature elsewhere emphasize notable differences not only in host-cell metabolism but also in immune responses when treated with viable Mtb compared to dead or iMtb. Therefore, it is recommended to incorporate viable Mtb in experiments where only iMtb was utilized. Also, in the abstract (3rd sentence), do the authors refer to live or irradiated Mtb? It is imperative to clearly indicate this distinction, as the subsequent conclusions are based only on one of these two scenarios, not both. The contradictory mitochondrial mass results (figure 1; live and dead Mtb showed opposite mitochondrial mass results) clearly illustrate the profound difference live (versus dead) Mtb cells can have on an experiment.

We thank the reviewer for stating this concern. For Figure 3, the involvement of TLR2 ligation on lactate release was also confirmed with live Mtb (shown in Figure S2D). In this current version, we also confirmed the involvement of TLR2 ligation in the up-regulation of HIF-1α triggered by live Mtb (new Fig S2E). As for Figure 4, we agree that performing assays with live Mtb will add complementary information. Indeed, we hope to investigate in the future the impact of the glycolysis/HIF1a axes on the adaptive immune response. We believe that employing live bacteria and considering their active immune evasion strategies will be crucial. However, at present, this is not the focus of the current manuscript and is beyond its scope.

We also agree with the reviewer that confirmation of the migratory behavior of DCs following Mtb infection is a crucial aspect of the study. To comply with this pertinent request, we performed new migration assays using Mtb-infected DCs treated with oxamate or PX-478 to validate that the HIF1a/glycolysis axis; results convincingly demonstrate that this axis is essential for DC migration, particularly in the context of Mtb-infected cells (new Fig 5D). Having observed the same inhibitory effect of HIF1a and LDH inhibition on cell migration in either Mtb-infected or iMtb-stimulated DCs, we consider that the sentence alluded to by the reviewer in the abstract is now applicable to both contexts (page 2, line 34-36). We hope this reviewer agrees.

(6) The discussion and the graphical abstract elucidating the distinctions in glycolysis between CD16+ monocytes of HS and TB patients and iMtb-treated Mo-DCs are currently confusing and require clarification. According to the abstract, monocytes from TB patients exhibit heightened glycolysis, resulting in diminished HIF-a activity and migratory capacity of MO-DCs. This prompts a question: if exacerbated glycolysis in monocytes is associated with adverse outcomes, wouldn't it be logical to consider suppressing glycolysis? If so, how can inhibiting glycolysis, a favored metabolic pathway for pro-inflammatory responses, be beneficial for TB therapy?

We understand the reviewer’s concern about this apparent paradox. As previously mentioned in response to the public review provided by the reviewer, inhibiting glycolysis may yield varying outcomes depending on the stage of infection, as well as the cellular target (e.g., monocytes, DCs) or compartment (systemic *versus* local). It is imperative to delve deeper into the potential role of the HIF1α/glycolysis axis at the systemic level within the context of chronic inflammation, contrasting with its role in a local setting during the acute phase of infection.

A comprehensive understanding of the interplay between Mtb infection and glycolysis in myeloid cells requires further consideration of various contextual conditions, urging caution against oversimplified interpretations. For instance, one of the objectives of host-directed therapies (HDTs) is to mitigate host-response inflammatory toxicity, which can impede treatment efficacy (doi: 10.3389/fimmu.2021.645485). In this regard, traditional anti-inflammatory drugs such as non-steroidal anti-inflammatory drugs (NSAIDs) and corticosteroids have been explored as adjunct therapies due to their immunomodulatory properties. Additionally, compounds like vitamin D, phenylbutyrate (PBA), metformin, and thalidomide, among others, have been investigated in the context of TB infections (doi:10.3389/fimmu.2017.00772), highlighting the diverse range of strategies aimed at enhancing TB treatment. These efforts extend beyond bolstering antimicrobial activity to encompass minimizing inflammation and mitigating tissue damage.

(7) I am not convinced that BubbleMap made any significant contribution to the manuscript perhaps because it is poorly described in the figure legends/main text (I am unable to determine what data set is significant or not).

We agree with the reviewer’s comment. To clarify the valuable information gleaned from these analyses, we have added interpretive guidelines on bubble color, bubble size and statistical significance in the legend of Figure 7. We hope these changes may reflect the significant contribution of the BubbleMap analysis approach to this study, which demonstrates a significant enrichment of interferon response gene expression in the monocyte compartment from patients with active TB compared to their control counterparts. Notably, this enrichment does not extend to genes associated with the OXPHOS hallmark.

(8) The use of cells/monocytes from TB patients is a concern in addition to the incomplete demographic table. In the case of the latter, absolute numbers including percentages should be included. Importantly, it appears that cells from TB patients were used, that received anti-TB drug therapy (regimen not stated) up to two weeks post diagnosis and not at baseline. This is important as recent studies have shown that anti-TB drugs modulates the bioenergetics of host cells. Lastly, what were the precise TB symptoms the authors referred to in figure 7C?

We have updated the demographic table and included the absolute numbers. We concur with the reviewer's viewpoint, particularly in light of recent findings illustrating the impact of anti-TB drug treatment on cell metabolism (doi: 10.1128/AAC.00932-21/). Again, this study underscores the complexity of such effects, which exhibit considerable variability influenced by factors such as cell type, drug concentration, and combination therapy.

Despite this variability, our analysis involving monocytes from TB patients, who received different antibiotic combinations within short time frames (less than 15 days) reveals a marked increase in glycolysis in CD16+ monocytes compared to healthy counterparts. We did not observe a correlation between monocyte glycolytic capacity and the start time of antibiotic treatment within this 15-day window (see below, Author response image 1). These findings suggest that the antibiotic regimen does not have a significant impact on monocyte glycolytic capacity during the first 15 days. However, we did observe an effect of antibiotic treatment when comparing patients before and 2 months after treatment. Enrichment analysis of various monocyte subsets before and after 2 months of treatment (GEO accession number: GSE185372) showed that CD14dim CD16+ and CD14+ CD16+ populations had higher glycolytic activity before treatment, which is decreased then post-treatment (Author response image 2).

**Author response image 1. sa2fig1:** Correlation analysis between the baseline glycolytic capacity and the time since treatment onset for each monocyte subset (CD14+CD16-, CD14+CD16+ and CD14dimCD16+, N = 11). Linear regression lines are shown. Spearman’s rank test. The data are represented as scatter plots with each circle representing a single individual.

**Author response image 2. sa2fig2:** Gene enrichment analysis for glycolytic genes on the pairwise comparisons of each monocyte subset (CD14+CD16-, CD14+CD16+ and CD14dimCD16+) from patients with active TB pre-treatment vs patients with active TB (TB) undergoing treatment for 2 months. Comparisons with a p-value of less than 0.05 and an FDR value of less than 0.25 are considered significantly different.

Overall, our results indicate that while drug treatment does affect cell bioenergetics, this effect is not prominent within the first 15 days of treatment. CD16+ monocytes maintain high basal glycolytic activity that normalizes after treatment, contrasting with the CD16- population (even under the same circulating antibiotic doses). This highlights the intricate interplay between anti-TB drugs and cellular metabolism, underscoring the need for further research to understand the underlying mechanisms and therapeutic implications.

Finally, the term symptoms evolution refers to the time period during which a patient experiences cough and phlegm for more than 2-3 weeks, with or without sputum that may (or not) be bloody, accompanied by symptoms of constitutional illness (*e.g,* loss of appetite, weight loss, night sweats, general malaise). As requested, this definition has been included in the method section (page 28-29, lines 705-709).

Minor:(1) Incorporate the abbreviation for tuberculosis "(TB)" in the first line of the abstract and similarly introduce the abbreviation for Mycobacterium tuberculosis when it is first mentioned in the abstract.

Thank you, we have amended it accordingly.

(2) As the majority of experiments are in vitro, the authors should specify the number of times each experiment was conducted for every figure.

We have included this information in each figure legend (see N for each panel). Since the majority of our approaches are conducted in vitro using primary cell cultures (specifically, human monocyte-derived DCs), we utilized samples from four to ten independent donors, not replicates, in order to account for the variability seen between donors.

(3) Rename Fig-2. Ensure consistent labeling for the metabolic dependency of uninfected, Mtb-infected, and the Bystander panel, aligning with the format used in panels A & B. Similarly, replace '-' with 'uninfected'.

We have modified the figure following most of the reviewer’s suggestions. However, we decided to keep the nomenclature “-” to denote a control condition, which can be unstimulated (panels A-B, fig 2) or uninfected cells (panels C-D, fig 2) depending on the experimental design.

(4) Discussion: It is unclear what the authors mean by 'some sort of exhausted glycolytic capacity'.

We have slightly modified the phrase.